# Accelerated phosphatidylcholine turnover in macrophages promotes adipose tissue inflammation in obesity

Kasparas Petkevicius[1†]*, Sam Virtue[1], Guillaume Bidault[1], Benjamin Jenkins[1], Cankut Çubuk[2,3,4], Cecilia Morgantini[5], Myriam Aouadi[5], Joaquin Dopazo[2,3,4], Mireille J Serlie[6], Albert Koulman[1], Antonio Vidal-Puig[1,7]*

[1]University of Cambridge Metabolic Research Laboratories, Institute of Metabolic Science, MDU MRC, Cambridge, United Kingdom; [2]Clinical Bioinformatics Area, Fundación Progreso y Salud, CDCA, Hospital Virgen del Rocio, Sevilla, Spain; [3]Functional Genomics Node, INB-ELIXIR-es, FPS, Hospital Virgen del Rocio, Sevilla, Spain; [4]Bioinformatics in Rare Diseases (BiER), Centro de Investigación Biomédica en Red de Enfermedades Raras (CIBERER), FPS, Hospital Virgen del Rocio, Sevilla, Spain; [5]Department of Medicine, Integrated Cardio Metabolic Centre, Karolinska Institutet, Huddinge, Sweden; [6]Department of Endocrinology and Metabolism, Amsterdam University Medical Center, Amsterdam, Netherlands; [7]Wellcome Trust Sanger Institute, Hinxton, United Kingdom

*For correspondence:
kp416@medschl.cam.ac.uk (KP);
ajv22@medschl.cam.ac.uk (AV-P)

Present address: [†]Diabetes Bioscience, Cardiovascular, Renal and Metabolism, IMED Biotech Unit, AstraZeneca, Gothenburg, Sweden

Competing interests: The authors declare that no competing interests exist.

**Abstract** White adipose tissue (WAT) inflammation contributes to the development of insulin resistance in obesity. While the role of adipose tissue macrophage (ATM) pro-inflammatory signalling in the development of insulin resistance has been established, it is less clear how WAT inflammation is initiated. Here, we show that ATMs isolated from obese mice and humans exhibit markers of increased rate of de novo phosphatidylcholine (PC) biosynthesis. Macrophage-specific knockout of phosphocholine cytidylyltransferase A (CCTα), the rate-limiting enzyme of de novo PC biosynthesis pathway, alleviated obesity-induced WAT inflammation and insulin resistance. Mechanistically, CCTα-deficient macrophages showed reduced ER stress and inflammation in response to palmitate. Surprisingly, this was not due to lower exogenous palmitate incorporation into cellular PCs. Instead, CCTα-null macrophages had lower membrane PC turnover, leading to elevated membrane polyunsaturated fatty acid levels that negated the pro-inflammatory effects of palmitate. Our results reveal a causal link between obesity-associated increase in de novo PC synthesis, accelerated PC turnover and pro-inflammatory activation of ATMs.
DOI: https://doi.org/10.7554/eLife.47990.001

## Introduction

Obesity-related metabolic disorders are among the most prevalent causes of death worldwide. Secondary complications of obesity have been suggested to be caused by the functional failure of white adipose tissue (WAT), leading to ectopic lipid deposition, lipotoxicity and systemic insulin resistance (*Virtue and Vidal-Puig, 2010*). Obesity is associated with a chronic low-grade inflammation, characterised by immune cell infiltration to WAT, a switch of adipose tissue macrophage (ATM) polarisation from a tissue-remodelling (M2) to a pro-inflammatory (M1) state and elevated production of pro-inflammatory, insulin-desensitising cytokines, such as tumour necrosis factor α (TNFα). Over the last decade, multiple genetic and pharmacological approaches have defined a causal role of macrophage-driven WAT inflammation in the development of insulin resistance (*Hotamisligil, 2017*).

**eLife digest** Although inflammation can be good for the body and help fight off infection, in certain cases it can also be harmful. When immune cells switch on at the wrong time, they can cause damage to cells and tissues. Fat tissue has its own population of immune cells called adipose tissue macrophages that remove dead fat cells and keep the tissue working. However, obesity changes the behaviour of these macrophages so they switch on as though they were fighting an infection and make the fat tissue inflamed. The signals produced by these activated macrophages stop fat tissue working, and this can lead to type 2 diabetes.

The trigger that activates macrophages in obesity is not yet clear, but some evidence suggests that it is due to the type of fat available. Fats come in two main forms: saturated, which can lead to high cholesterol, or unsaturated which can reduce the risk of high blood pressure. An increase in saturated fats can cause cells, including macrophages, to become stressed.

Researchers showed in 2011 that macrophages in the fatty tissue of obese mice accumulate fat and become inflamed, but it was unclear which types of fat, if any, were driving the inflammation.

Now, Petkevicius et al. – including some of the researchers involved in the 2011 work – report that macrophages in the fatty tissue of obese mice make excess phosphatidylcholine, a fat normally found in the cell membrane. Phosphatidylcholine is a type of fat known as a phospholipid and it is made up of two subunits called fatty acids that can either be saturated or unsaturated. In obese people and mice, fatty tissue produces too much of the enzyme that makes phosphatidylcholine, called CCTa.

Petkevicius et al. showed that partially removing the CCTa gene from macrophages reduces inflammation, but, unexpectedly, the amount of phosphatidylcholine in the cells stays the same. This is because macrophages respond to the halt in phosphatidylcholine production by removing less of the phospholipid from the membrane. This gives the macrophages time to exchange the saturated fatty acids in phosphatidylcholine for unsaturated fatty acids. Therefore, the longer phosphatidylcholine stays in the membrane, the more likely it is to contain unsaturated fatty acids. Further experiments demonstrated that this change counteracts the effects caused by excess saturated fats, protecting the cells and reducing inflammation.

Although the understanding of obesity is still in its early stages, this study adds another piece of the puzzle. If we can understand why fat stops working in obesity, and how this leads to disease, it could aid the design of new treatments for type 2 diabetes.

DOI: https://doi.org/10.7554/eLife.47990.002

However, specific pathophysiological mechanisms triggering pro-inflammatory activation of ATMs during obesity are poorly understood.

Our previous work identified that the lipid composition of ATMs undergoes both quantitative and qualitative changes during obesity (*Prieur et al., 2011*). Qualitative changes in the lipid composition of both plasma and endoplasmic reticulum (ER) membranes represent a major factor promoting insulin resistance (*Fu et al., 2011*; *Holzer et al., 2011*; *Wei et al., 2016*). Obesity-associated alterations in ER lipid composition lead to a cellular process termed ER stress, which invokes an adaptive unfolded protein response (UPR) (*Hou et al., 2014*). In macrophages, the UPR is coupled to the activation of intracellular inflammatory signalling pathways that cause WAT inflammation and insulin resistance (*Robblee et al., 2016*; *Shan et al., 2017*; *Suzuki et al., 2017*). Furthermore, M1 macrophages are characterised by increased endogenous fatty acid synthesis, which stabilises lipid rafts within plasma membrane to allow pro-inflammatory signal transduction in obesity (*Wei et al., 2016*). While phospholipids (PLs) are the main constituents of plasma and ER membranes, the importance of PL biosynthesis in ATMs during obesity has not yet been investigated.

The concept that macrophage ER stress could be induced during obesity due to changes in membrane composition is in line with the known physiological changes in lipid metabolism that occur during obesity. Obesity is associated with increased circulating saturated fatty acids (SFAs), which cause cellular ER stress by being incorporated into membrane PLs, leading to a decreased membrane fluidity due to increased membrane PL acyl chain saturation. Increased SFA-mediated ER rigidification is directly sensed by the transmembrane domains of UPR-transducing proteins (*Robblee et al.,*

*2016*; *Volmer et al., 2013*). ER saturation and the resulting UPR can be counteracted by both endogenously and exogenously derived mono- and polyunsaturated fatty acids (MUFAs and PUFAs), and PUFA-containing phospholipids, in particular phosphatidylcholines (PCs) (*Ariyama et al., 2010*; *Gianfrancesco et al., 2019*; *Robblee et al., 2016*; *Rong et al., 2013*).

PC is the most abundant PL in mammalian cells. Most cells can synthesise PC de novo through the Kennedy pathway, involving the transfer of phosphocholine onto diacylglycerol moiety. PCs synthesised de novo predominantly contain saturated (SFAs) and monounsaturated fatty acids (MUFAs), while polyunsaturated fatty acids (PUFAs) are incorporated into PCs via the Lands cycle, involving a hydrolysis of a single fatty acyl chain and esterification of a free PUFA to a resulting lysophosphatidylcholine (lysoPC) (*Shindou et al., 2013*). Metabolic flux through the de novo PC synthesis pathway and cellular PC levels are greatly increased in differentiating macrophages (*Ecker et al., 2010*). Furthermore, pro-inflammatory signalling via toll-like receptor 4 (TLR4) increases the rate of choline uptake and de novo PC synthesis in macrophages (*Sanchez-Lopez et al., 2019*; *Snider et al., 2018*; *Tian et al., 2008*). However, de novo PC synthesis in mature macrophages is not coupled to the expansion of the cellular PC pool, as it is counteracted by phospholipase D activity, leading to a rapid turnover of membrane PCs (*Jackowski et al., 1997*). The consequences of altered PC turnover in metabolic disease are not currently known.

Conceptually, the rate of de novo PC synthesis and turnover should affect PC remodelling via Lands cycle. The role of the Lands cycle in ER stress function has been studied by genetic manipulation of the enzyme lysoPC-acyltransferase 3 (LPCAT3), the major LPCAT isoform in macrophages (*Jiang et al., 2018*). LPCAT3 overexpression increases the rate of PUFA incorporation into PCs and protects cells from palmitate-induced ER stress, while the loss of LPCAT3 sensitises cells to palmitate lipotoxicity (*Rong et al., 2013*).

We have previously shown that during obesity, ATMs acquire an M1 phenotype concomitantly with their intracellular lipid accumulation (*Prieur et al., 2011*). Here, we demonstrate that markers of de novo PC synthesis are increased in ATMs isolated from obese mice and humans. *Lep*^ob/ob^ mice with a myeloid cell-specific reduction in de novo PC synthesis rate display reduced adipose tissue inflammation and improved metabolic profile compared to controls. Mechanistically, we show that reducing the activity of the de novo PC synthesis pathway by 30% does not reduce total cellular PC levels in macrophages. Instead, the reduction in PC synthesis is balanced by a reduction in PC degradation, maintaining the cellular PC pool size but increasing the half-life of PCs. The extended PC half-life leads to increased incorporation of PUFAs into PCs by allowing more time for PC remodelling. Elevated PC PUFA content protects macrophages from palmitate-induced ER stress and pro-inflammatory activation.

## Results

### Obesity accelerates de novo PC biosynthesis in ATMs

In order to identify intrinsic metabolic pathways associated with the phenotypic switch of ATMs towards an M1 polarisation state, we reanalysed our published microarray dataset (GSE36669) of epididymal WAT (eWAT) macrophages isolated from WT and *Lep*^ob/ob^ animals using inferred metabolic flux analysis (*Cubuk et al., 2018*; *Çubuk et al., 2019*). We focused on the pathways that were unchanged or downregulated in 5-week-old *Lep*^ob/ob^ ATMs, which are predominantly M2-polarised, but upregulated at 16 weeks of age, when eWAT of *Lep*^ob/ob^ mice is inflamed (*Prieur et al., 2011*). Among the metabolic pathways that fitted these criteria was de novo PC biosynthesis (*Figure 1a–b*), with a lower inferred activity score in 5-week-old, but higher score in 16-week-old *Lep*^ob/ob^ ATMs compared to age-matched WT controls (*Figure 1d*). Further analysis of the processes that were unchanged at 5 weeks but upregulated at 16 weeks in *Lep*^ob/ob^ ATMs revealed several pathways related to PL metabolism (*Figure 1d*, *Supplementary file 1*). The activity of de novo PE biosynthesis pathway was not modulated in *Lep*^ob/ob^ ATMs (*Figure 1—figure supplement 1a–b*).

In order to determine whether increased inferred activity of the de novo PC synthesis pathway in obesity was specific to ATMs, or also occurred in other tissue-resident macrophages, we performed global transcriptomic comparison between liver macrophages isolated from 14-week-old *Lep*^ob/ob^ and control mice. Unlike ATMs, liver macrophages isolated from obese mice showed similar

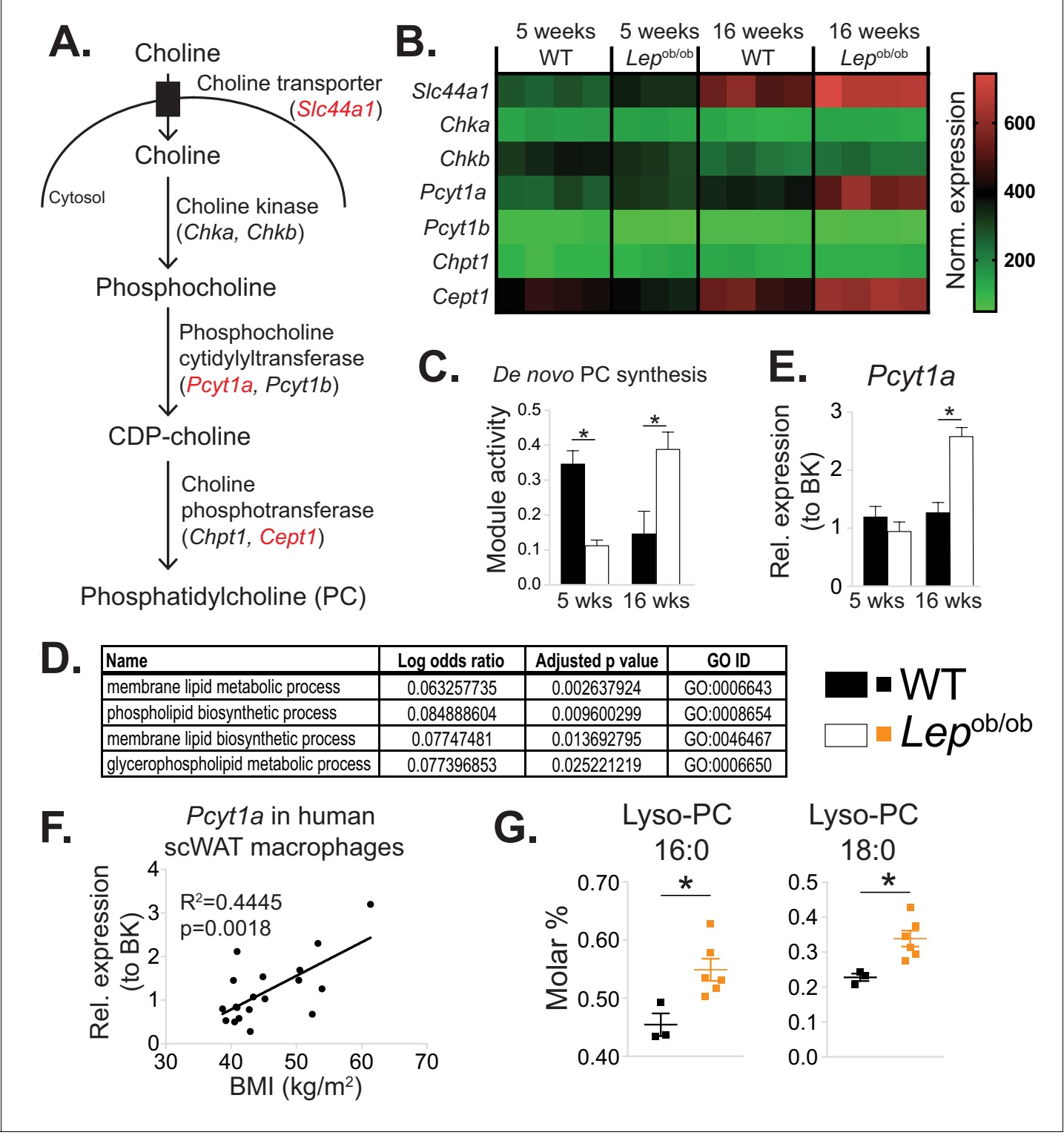

**Figure 1.** De novo PC synthesis rate is increased in ATMs during obesity. (A) Simplified schema of the Kennedy pathway of de novo PC biosynthesis. Transcripts in red are upregulated in 16-week-old $Lep^{ob/ob}$ eWAT macrophages compared to WT controls. (B) Normalised microarray gene expression values for the enzymes of the Kennedy pathway of de novo PC biosynthesis in eWAT macrophages. (C) De novo PC biosynthesis pathway module (M00090) activity in eWAT macrophages, as inferred by the Metabolizer algorithm. (D) Gene Ontology (GO) pathways related to membrane lipid metabolism that are increased in $Lep^{ob/ob}$ eWAT macrophages at 16 weeks, but not at 5 weeks of age compared to WT controls. (E) $Pcyt1a$ expression in eWAT macrophages, measured by qPCR. (F)$Pcyt1a$ expression, measured by qPCR in subcutaneous WAT macrophages isolated from obese patients

*Figure 1 continued on next page*

Figure 1 continued

undergoing bariatric surgery, plotted against their body weight (n = 19). (**G**) Molar abundance of 16:0 and 18:0 lyso-PC species (expressed as percentage of total measured PLs) in eWAT macrophages (n = 3 pools of 5 WT, n = 6 $Lep^{ob/ob}$ mice). *p<0.05 between genotypes, error bars indicate SEM.

DOI: https://doi.org/10.7554/eLife.47990.003

The following figure supplements are available for figure 1:

**Figure supplement 1.** Obesity does not affect de novo PE synthesis rate in ATMs.

DOI: https://doi.org/10.7554/eLife.47990.004

**Figure supplement 2.** Obesity does not affect de novo PC synthesis rate in liver macrophages.

DOI: https://doi.org/10.7554/eLife.47990.005

**Figure supplement 3.** *Pcyt1a* expression levels in different tissue macrophage populations.

DOI: https://doi.org/10.7554/eLife.47990.006

**Figure supplement 4.** Obesity tends to increase PC molar percentage and PC to PE molar ratio in ATMs.

DOI: https://doi.org/10.7554/eLife.47990.007

**Figure supplement 5.** The effects of obesity on PC remodelling gene expression in ATMs.

DOI: https://doi.org/10.7554/eLife.47990.008

expression levels of de novo PC biosynthesis pathway constituents compared to controls (*Figure 1— figure supplement 2*).

In accordance with our ATM transcriptomic analysis, the expression of *Pcyt1a*, encoding phosphocholine cytidylyltransferase A (CCTα), the rate-limiting enzyme in de novo PC synthesis pathway, was unchanged at 5 weeks, but increased at 16 weeks in $Lep^{ob/ob}$ ATMs compared to WT controls when measured by qPCR (*Figure 1e*). In contrast, the *Pcyt1a* paralogue *Pcyt1b* was down-regulated at 5 weeks and not modulated at 16 weeks in $Lep^{ob/ob}$ ATMs (*Figure 1—figure supplement 1c*). Furthermore, *Pcyt1a* expression in macrophages isolated from the WAT of obese individuals was positively correlated with BMI (*Figure 1f*). Of note, out of all analysed tissue macrophage populations publicly available in Immgen database (*Heng et al., 2008*), ATMs had the highest expression of *Pcyt1a* transcript (*Figure 1—figure supplement 3*).

Next, we reanalysed our previously published lipid profiles from $Lep^{ob/ob}$ ATMs (*Prieur et al., 2011*), focusing only on measured PL species. Relative to the total PL amount, both PC abundance and PC:PE molar ratio tended to increase (*Figure 1—figure supplement 4a–b*), and palmitate- and stearate-containing lysoPC species were upregulated in 16-week-old $Lep^{ob/ob}$ ATMs compared to WT controls (*Figure 1g*). It has been previously shown that hepatic lysoPC levels are reduced when the balance between PC synthesis and LPCAT activity is perturbed. Specifically, increasing LPCAT3 activity without changing de novo PC synthesis or breakdown reduces LysoPC levels, as LPCAT3 re-esterifies LysoPC into PC (*Rong et al., 2013*). Conversely, in obese WAT, despite the upregulation of *Lpcat3* transcript at both 5 and 16 weeks in $Lep^{ob/ob}$ ATMs (*Figure 1—figure supplement 5a*), we observed increased lysoPC species in $Lep^{ob/ob}$ ATMs (*Figure 1g*). The elevation in LysoPCs was therefore consistent with obesity causing a disproportional increase in the rate of both de novo PC synthesis and hydrolysis that exceeded the capacity of LPCAT3 to re-esterify Lyso-PC back into PC (*Figure 1—figure supplement 5b*).

## *Pcyt1a* deletion in myeloid cells improves glucose metabolism in obese mice

To test if increased PC biosynthesis in ATMs affected whole-organism metabolic homeostasis, we investigated mice with *Pcyt1a* deletion in myeloid cells (CCTα mKO) that have been described previously (*Tian et al., 2008*). Initially, we sought to validate whether the loss of *Pcyt1a* would impact macrophage differentiation or function in vitro and in vivo. As indicated by the normal surface expression of macrophage markers F4/80, CD206 and CD301, unaltered bacterial phagocytosis and normal TNFα and IL-6 cytokine secretion in response to LPS, the differentiation of CCTα-null bone marrow cells into macrophages (BMDMs) was not impaired (*Figure 2—figure supplement 1a–c*). *Pcyt1a* transcript levels in BMDMs on a C57Bl/6J genetic background were reduced by ~50%, which translated into ~80% decrease in CCTα protein expression and ~30% decrease in de novo PC synthesis rate compared to controls (*Figure 2—figure supplement 1d*). In vivo, the expression of macrophage mRNA markers in eWAT and liver was comparable between CCTα mKO and control

animals (*Figure 2—figure supplement 2a–b*). Overall, these results confirmed that loss of CCTα reduced de novo PC biosynthesis rate in macrophages without altering their development or function.

CCTα mKO mice exhibited similar growth rates and metabolic tissue weights compared to controls (*Figure 2—figure supplement 3a–b*). No differences in glucose or insulin tolerance tests were observed between CCTα mKO and control groups (*Figure 2—figure supplement 3c–d*). In accordance, the expression levels of insulin-regulated metabolic genes were similar in eWAT and liver of CCTα mKO and control mice (*Figure 2—figure supplement 2a–b*).

We next evaluated the importance of increased macrophage de novo PC synthesis in obesity. We first confirmed that bone marrow transplantation did not alter the induction of *Pcyt1a* in the *Lep*ob/ob genetic background (*Figure 2—figure supplement 4*). Next, we transplanted CCTα mKO or control bone marrow into irradiated *Lep*ob/ob animals (*Figure 2a*). While no differences in post-irradiation body weight gain, WAT and liver mass were observed (*Figure 2b–c*), *Lep*ob/ob mice carrying CCTα mKO bone marrow tended to have improved glucose tolerance and exhibited increased sensitivity to exogenous insulin compared to controls (*Figure 2d–e*). Overall, macrophage-specific *Pcyt1a* deletion did not affect ATM development, adipose tissue function and glucose metabolism in lean animals, but improved systemic glucose handling in *Lep*ob/ob mice, the model of obesity in which we originally observed an induction of *Pcyt1a* in the ATM population.

## *Pcyt1a* deletion in myeloid cells alleviates inflammation and improves insulin signalling in the WAT of obese mice

While the metabolic effects of macrophage-specific *Pcyt1a* deletion on a *Lep*ob/ob background were modest, they were consistent with the relatively small reduction (30%) in de novo PC biosynthesis rate we observed in BMDMs in vitro. We next sought to determine how *Pcyt1a* deficiency in macrophages improved glucose metabolism in obese mice. First, we performed whole transcriptome comparison of eWAT isolated from CCTα mKO *Lep*ob/ob BMT and control animals. Pathway analysis of the transcriptomic data revealed an increase in transcripts associated with insulin sensitivity and glucose metabolism, while pathways related to ER stress and macrophage-driven inflammation were supressed in CCTα mKO compared to control BMT *Lep*ob/ob mice (*Figure 3a*, *Supplementary files 2a-b*). RNA sequencing results were also confirmed by qPCR (*Figure 2—figure supplement 1a*). In accordance to gene expression data, insulin-responsive AKT phosphorylation was increased in the eWAT of CCTα mKO *Lep*ob/ob animals compared to controls (*Figure 3b–c*). Furthermore, while we found no differences in total ATM number, eWAT macrophages showed a shift from M1 to M2 polarisation in CCTα mKO compared to controls (*Figure 3d*). No differences in the number of crown-like structures (CLS) and eWAT adipocyte area were observed between genotypes (*Figure 3—figure supplement 2a–c*). Finally, unlike eWAT, the expression of pro-inflammatory and insulin-responsive marker genes in the liver were similar between genotypes (*Figure 2—figure supplement 1b*). Similarly, insulin-responsive AKT phosphorylation was comparable between the livers of both genotypes (*Figure 3—figure supplement 3a–b*) and only tended to be increased in the skeletal muscles of CCTα mKO compared to control BMT *Lep*ob/ob mice (*Figure 3—figure supplement 3c–d*), indicating that macrophage-specific *Pcyt1a* deletion had a stronger impact on white adipose tissue insulin signalling, compared to liver and muscle. Altogether, we have found that reducing de novo PC biosynthesis rate in macrophages alleviates WAT inflammation and insulin resistance in obese mice, without affecting total ATM and CLS number.

## Loss of *Pcyt1a* lowers palmitate-induced ER stress and inflammation in macrophages

Next, we sought to investigate the molecular events that reduce WAT inflammation in obese animals carrying CCTα mKO bone marrow. For this purpose, we utilised an in vitro model of BMDMs exposed to high palmitate concentrations. We selected palmitate concentrations that have previously been reported to induce ER stress and pro-inflammatory activation, thus mimicking the effects of obesity on ATMs (*Robblee et al., 2016*). We observed diminished *Tnf* transcript levels in palmitate-treated CCTα-null macrophages compared to controls (*Figure 4a*). Reduced inflammation in BMDMs was accompanied by a lower ER stress response to palmitate, as indicated by lower

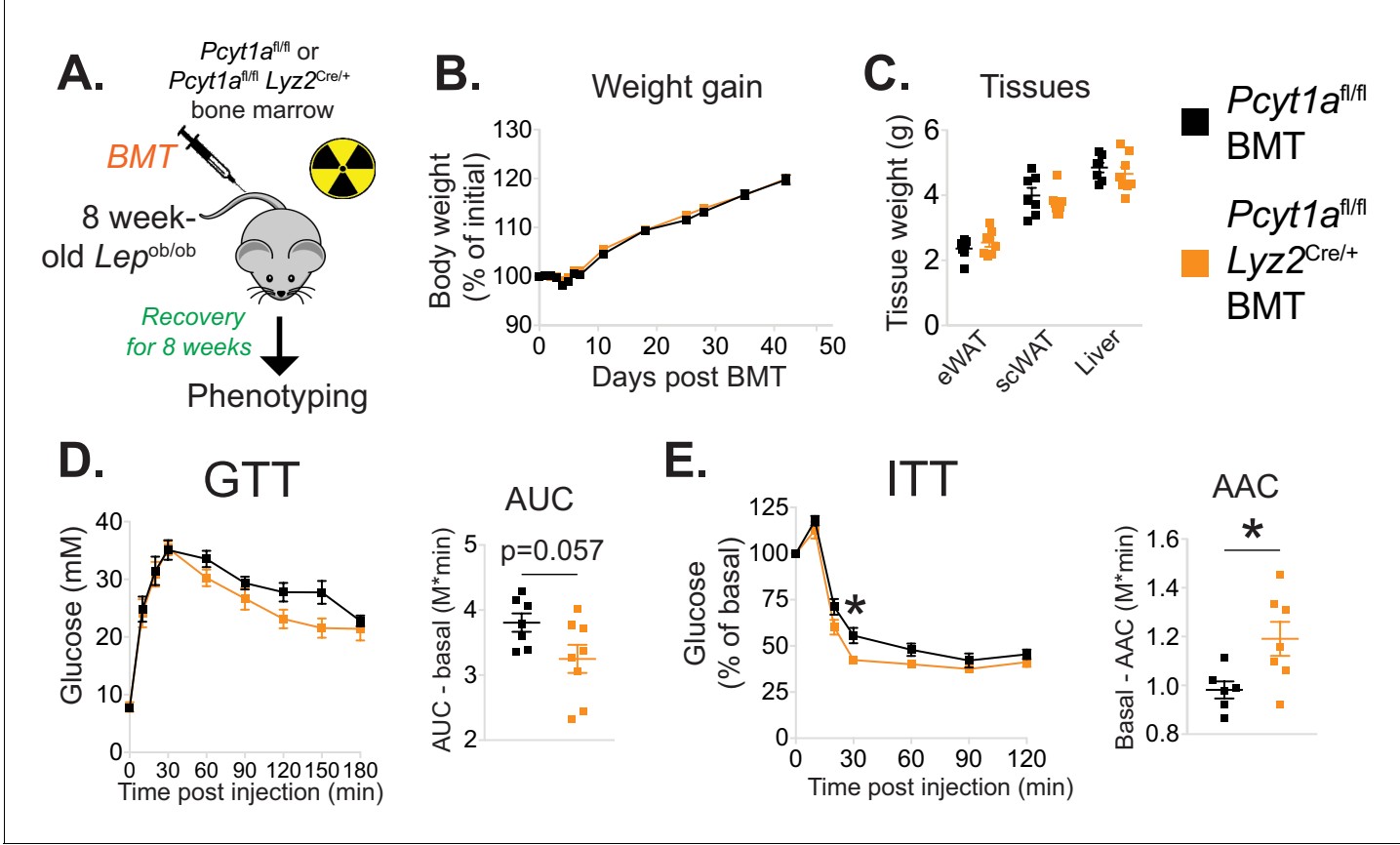

**Figure 2.** Myeloid cell-specific deletion of *Pcyt1a* leads to improved systemic glucose metabolism on the $Lep^{ob/ob}$ genetic background. (**A**) Schema of the BMT study design. (**B**) Body weight gain curves and (**C**) weights of indicated tissues of $Lep^{ob/ob}$ mice transplanted with $Pcyt1a^{fl/fl}$ (n = 7) or $Pcyt1a^{fl/fl}$ $Lyz2^{Cre/+}$ (n = 8) bone marrow. (**D**)GTT curves and areas under curve (AUC), normalised to basal glucose levels. (**E**) ITT curves, presented as percentage values of basal glucose levels, and areas above curve (AAC), normalised to basal glucose levels.

DOI: https://doi.org/10.7554/eLife.47990.009

The following figure supplements are available for figure 2:

**Figure supplement 1.** Myeloid cell-specific deletion of *Pcyt1a* does not impair BMDM differentiation or function in vitro.

DOI: https://doi.org/10.7554/eLife.47990.010

**Figure supplement 2.** Myeloid cell-specific deletion of *Pcyt1a* does not affect eWAT or liver gene expression in lean mice.

DOI: https://doi.org/10.7554/eLife.47990.011

**Figure supplement 3.** Myeloid cell-specific deletion of *Pcyt1a* does not affect growth or glucose metabolism of lean mice.

DOI: https://doi.org/10.7554/eLife.47990.012

**Figure supplement 4.** WT to $Lep^{ob/ob}$ bone marrow transplant does not affect the increase of *Pcyt1a* transcription in eWAT ATMs.

DOI: https://doi.org/10.7554/eLife.47990.013

induction of ER stress marker gene expression and reduced stress-responsive kinase activation in CCTα-null BMDMs compared to controls (***Figure 4b–d***).

Furthermore, CCTα-null BMDMs were less susceptible to palmitate-induced cell death than controls (***Figure 4—figure supplement 1a***). While *Pcyt1a* deficiency was protective against cytotoxicity in response to palmitate, it was detrimental in response to other ER stressors, including thapsigargin (***Figure 4—figure supplement 1b***) and free cholesterol (***Zhang et al., 2000***). Finally, cultured peritoneal macrophages isolated from CCTα mKO animals also showed reduced ER stress response to palmitate compared to controls (***Figure 4—figure supplement 2***). Overall, CCTα-null macrophages were protected against palmitate-induced ER stress and subsequent cytotoxicity and inflammation.

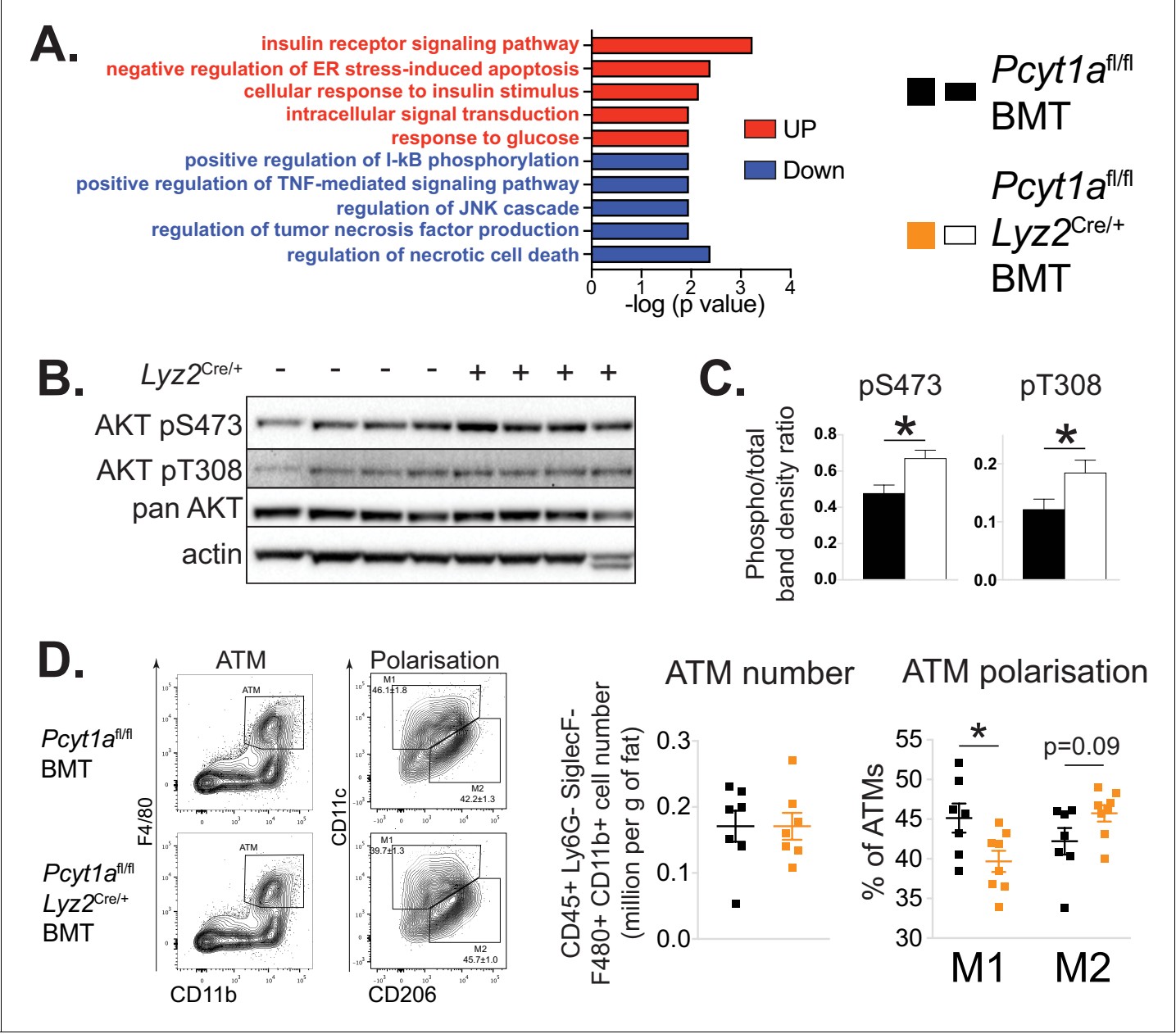

**Figure 3.** Myeloid cell-specific deletion of *Pcyt1a* leads to reduced WAT inflammation on the *Lep*ob/ob genetic background. (A) Selected pathways from eWAT RNAseq analysis, upregulated (red) or downregulated (blue) in *Lep*ob/ob mice transplanted with *Pcyt1a*fl/fl *Lyz2*Cre/+ (n = 8) bone marrow compared to controls (n = 7). (B) Representative AKT phosphorylation Western blots and (C) their densitometry quantification in eWAT of *Lep*ob/ob mice transplanted with *Pcyt1a*fl/fl (n = 7) or *Pcyt1a*fl/fl *Lyz2*Cre/+ (n = 8) bone marrow. (D) Flow cytometry gating strategy, quantification of ATM number per gram of eWAT and the relative polarisation of ATM population in *Lep*ob/ob mice.

DOI: https://doi.org/10.7554/eLife.47990.014

The following figure supplements are available for figure 3:

**Figure supplement 1.** The effects of myeloid cell-specific deletion of *Pcyt1a* on eWAT and liver gene expression on the *Lep*ob/ob genetic background.

DOI: https://doi.org/10.7554/eLife.47990.015

**Figure supplement 2.** Myeloid cell-specific deletion of *Pcyt1a* does not affect eWAT CLS number or adipocyte size on the *Lep*ob/ob genetic background.

DOI: https://doi.org/10.7554/eLife.47990.016

**Figure supplement 3.** The effects of myeloid cell-specific deletion of *Pcyt1a* on liver and skeletal muscle AKT signalling on the *Lep*ob/ob genetic background.

DOI: https://doi.org/10.7554/eLife.47990.017

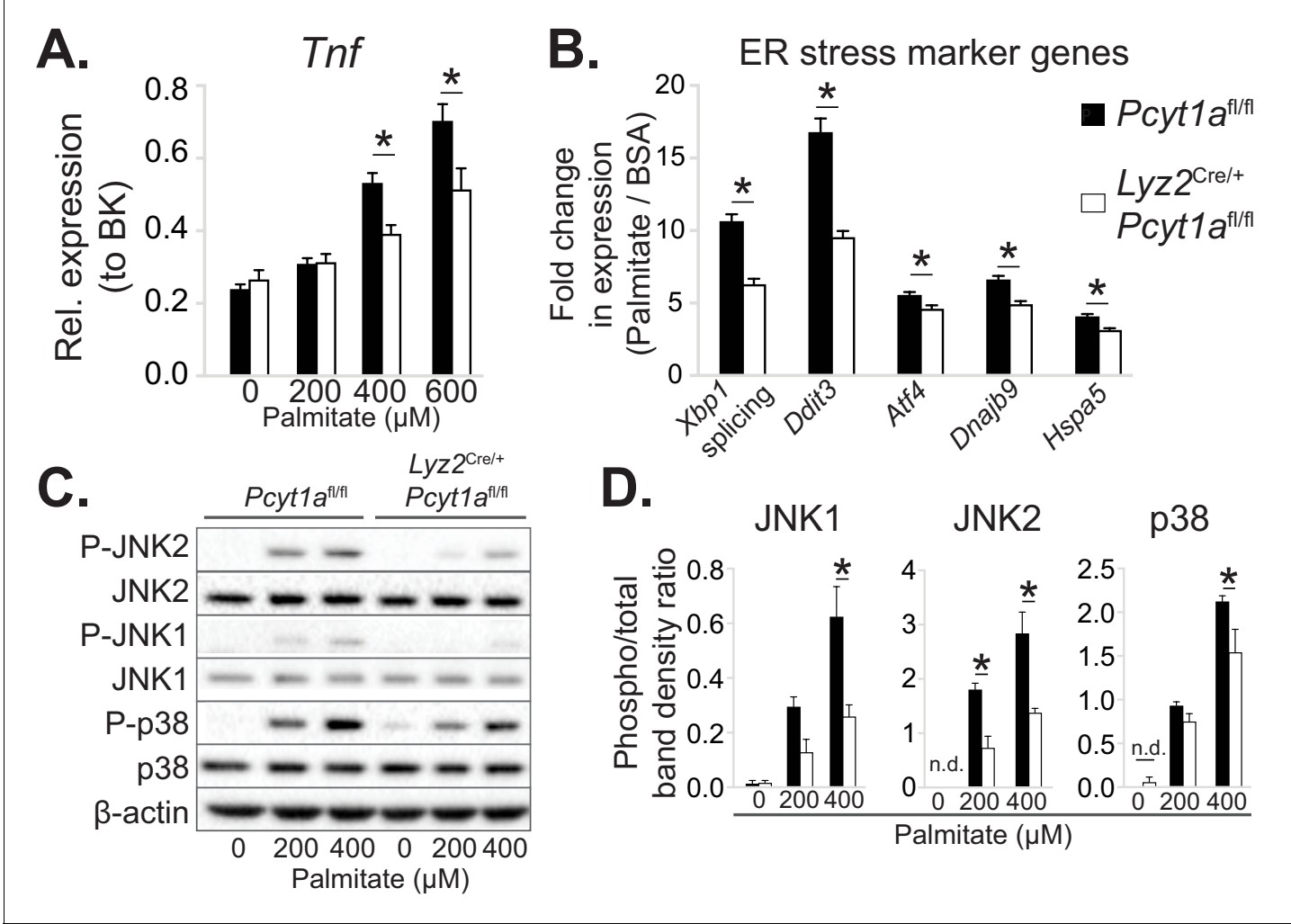

**Figure 4.** *Pcyt1a* deficiency protects macrophages from palmitate-induced ER stress and inflammation. (A) *Tnf* expression levels in *Pcyt1a*[fl/fl] (n=5) or *Pcyt1a*[fl/fl] *Lyz2*[Cre/+] (n=3) BMDMs treated with indicated doses of palmitate for 16 hours. (B) Fold induction (compared to BSA alone) of indicated ER stress marker gene expression *Pcyt1a*[fl/fl] (n=7) or *Pcyt1a*[fl/fl] *Lyz2*[Cre/+] (n=8) BMDMs treated with 250 µM palmitate for 16 hours. (C) Representative Western blots and (D) their densitometry quantification of *Pcyt1a*[fl/fl] (n=5) or *Pcyt1a*[fl/fl] *Lyz2*[Cre/+] (n=3) BMDMs treated with indicated doses of palmitate for 16 hours. *p < 0.05 between genotypes, error bars indicate SEM. All presented experiments are representative of at least 3 BMDM cultures.

DOI: https://doi.org/10.7554/eLife.47990.018

The following figure supplements are available for figure 4:

**Figure supplement 1.** *Pcyt1a* deficiency protects macrophages from palmitate, but not thapsigargin-induced cell death.

DOI: https://doi.org/10.7554/eLife.47990.019

**Figure supplement 2.** *Pcyt1a* deficiency protects peritoneal macrophages from palmitate-induced ER stress.

DOI: https://doi.org/10.7554/eLife.47990.020

### De novo PC biosynthesis pathway does not incorporate exogenous palmitate into membrane PCs

We next investigated how mitigating CCTα activity caused a reduction in palmitate-induced ER stress. As the de novo PC biosynthesis pathway had been suggested to control the flux of exogenous palmitate into cellular PCs (*Robblee et al., 2016*), we hypothesised that CCTα-null BMDMs would have a reduced rate of palmitate incorporation into their membranes. In order to test our hypothesis, we traced the incorporation of exogenous palmitate into cellular PCs over time. Surprisingly, CCTα-null and control BMDMs showed no differences in the rate of radiolabelled palmitate appearance in total lipid or PC fractions (*Figure 5a–b*).

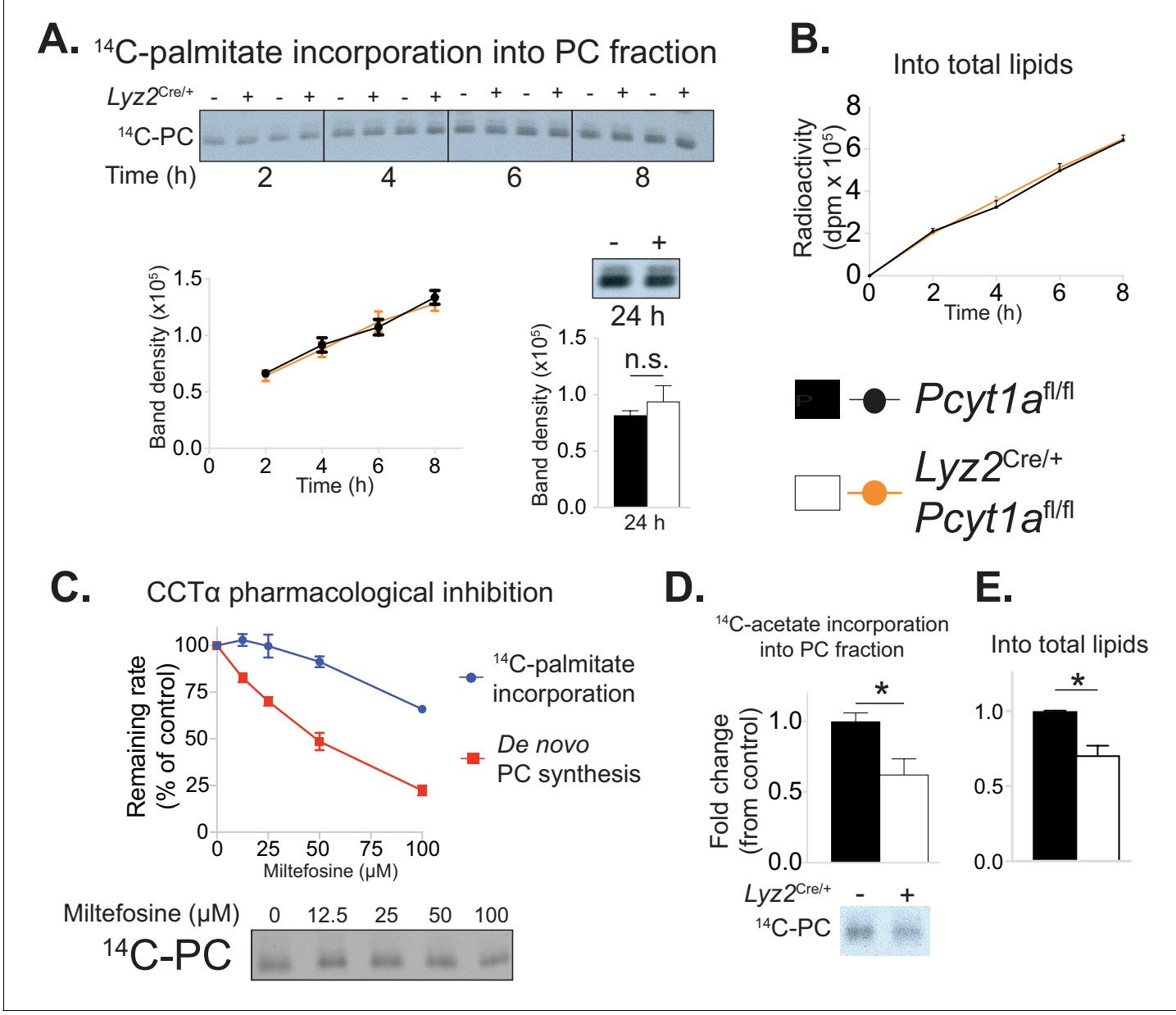

**Figure 5.** *Pcyt1a* deficiency in macrophages does not affect the rate of exogenous palmitate incorporation into PCs. (A) Representative autoradiogram and densitometry quantification of $^{14}$C-palmitate incorporation into PCs or (B) total lipids of *Pcyt1a*$^{fl/fl}$ (n = 4) or *Pcyt1a*$^{fl/fl}$ *Lyz2*$^{Cre/+}$ (n = 4) BMDMs treated with 250 μM palmitate for indicated periods of time. (C) Inhibition of de novo PC biosynthesis (red line) and $^{14}$C-palmitate incorporation into PC fraction (blue line) of WT BMDMs (n = 4), pretreated with indicated doses of miltefosine for 1 hr and stimulated with 250 μM palmitate for 3 hr. Representative autoradiogram is presented below. (D) Representative autoradiogram and densitometry quantification of $^{14}$C-acetate incorporation into PCs or (E) total lipids of untreated *Pcyt1a*$^{fl/fl}$ (n = 4) or *Pcyt1a*$^{fl/fl}$ *Lyz2*$^{Cre/+}$ (n = 4) BMDMs over 3 hr, normalised to *Pcyt1a*$^{fl/fl}$ group average. *p<0.05 between genotypes, error bars indicate SEM. All presented experiments are representative of at least 3 BMDM cultures.

DOI: https://doi.org/10.7554/eLife.47990.021

We then attempted to validate our unexpected findings using acute pharmacological inhibition of CCTα by miltefosine. Miltefosine reduced the rate of de novo PC synthesis in palmitate-treated BMDMs in a dose-response manner (*Figure 5c*). In contrast, only 100 μM concentration of miltefosine showed an inhibitory effect on the incorporation of palmitate into membrane PCs (*Figure 5c*). Importantly, and in line with the evidence from our genetic model, the dose of miltefosine that reduced de novo PC biosynthesis rate by 30% (as we have observed in CCTα-null BMDMs,

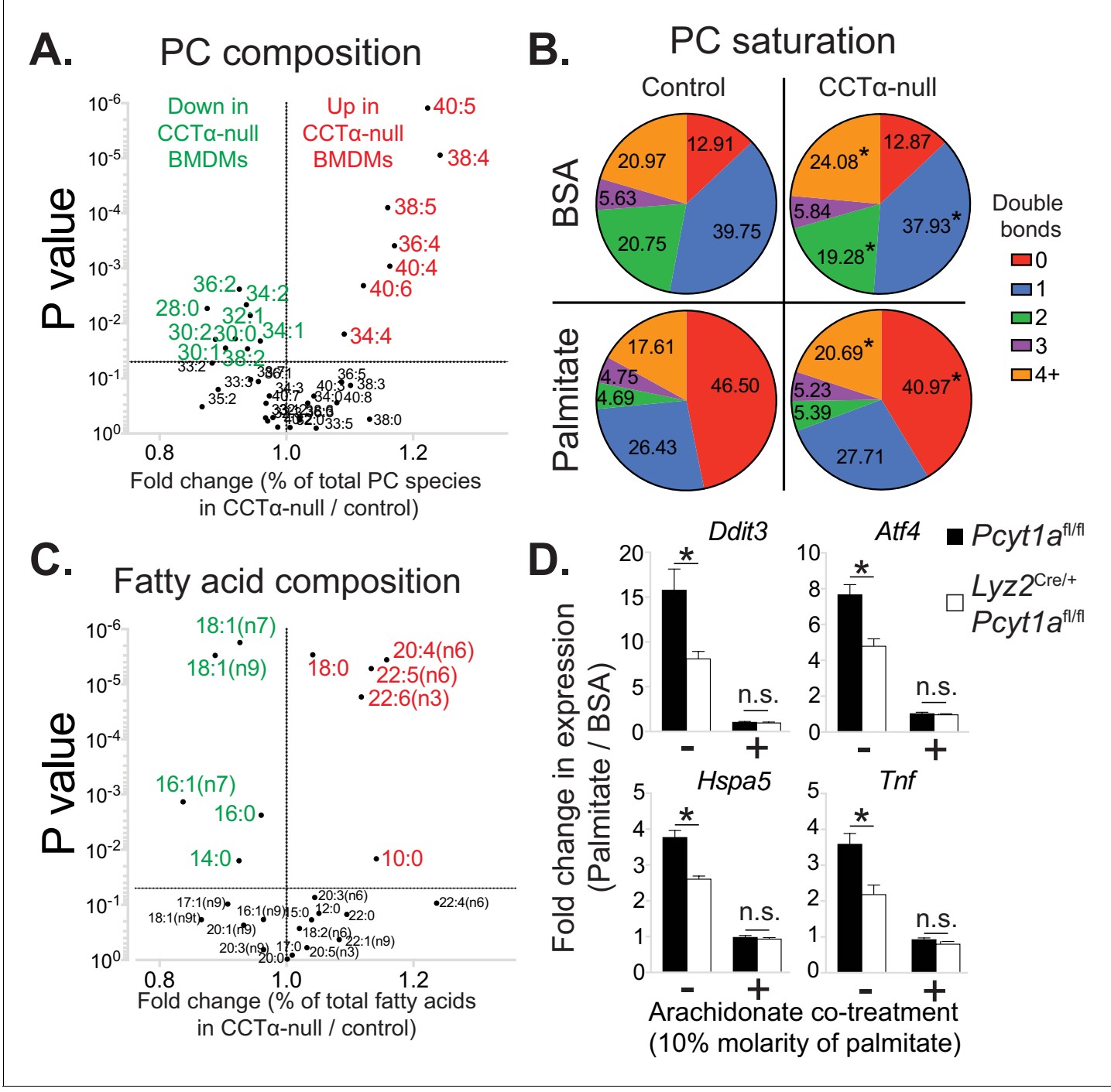

**Figure 6.** *Pcyt1a* deficiency increases PUFA-containing PC levels in macrophages. (**A**) Volcano plot of indicated PC species, expressed as molar percentage of all measured PCs, in *Pcyt1a*<sup>fl/fl</sup> (n = 7) or *Pcyt1a*<sup>fl/fl</sup> *Lyz2*<sup>Cre/+</sup> (n = 8) BMDMs. (**B**) Pie charts indicating the relative abundance of PC species with different degrees of unsaturation in *Pcyt1a*<sup>fl/fl</sup> (n = 7) or *Pcyt1a*<sup>fl/fl</sup> *Lyz2*<sup>Cre/+</sup> (n = 8) BMDMs in a basal state or after 16 hr treatment with 250 μM palmitate. (**C**) Volcano plot of indicated fatty acid species, expressed as molar percentage of all measured fatty acids, in *Pcyt1a*<sup>fl/fl</sup> (n = 7) or *Pcyt1a*<sup>fl/fl</sup> *Lyz2*<sup>Cre/+</sup> (n = 8) BMDMs. (**D**) Fold induction (compared to BSA alone) of indicated ER stress marker gene and *Tnf* expression in *Pcyt1a*<sup>fl/fl</sup> (n = 4) or *Pcyt1a*<sup>fl/fl</sup> *Lyz2*<sup>Cre/+</sup> (n = 4) BMDMs after 16 hr treatment with 250 μM palmitate, supplemented with or without 25 μM arachidonate.

DOI: https://doi.org/10.7554/eLife.47990.022

The following source data and figure supplements are available for figure 6:

**Source data 1.** A list of all measured lipid species by LC-MS in *Pcyt1a*<sup>fl/fl</sup> (n = 7) or *Pcyt1a*<sup>fl/fl</sup> *Lyz2*<sup>Cre/+</sup> (n = 8) BMDMs in a basal state or after 16 hr treatment with 250 μM palmitate.

*Figure 6 continued on next page*

*Figure 6 continued*

DOI: https://doi.org/10.7554/eLife.47990.027

**Figure supplement 1.** The effects of *Pcyt1a* deficiency on total PC and PE levels and PE composition in macrophages.

DOI: https://doi.org/10.7554/eLife.47990.023

**Figure supplement 2.** *Pcyt1a* deficiency reduces SREBP1 target gene expression in macrophages.

DOI: https://doi.org/10.7554/eLife.47990.024

**Figure supplement 3.** The effects of palmitate treatment on PC to PE ratio and PC composition in macrophages.

DOI: https://doi.org/10.7554/eLife.47990.025

**Figure supplement 4.** *Pcyt1a* deficiency increases total PUFA levels in macrophages.

DOI: https://doi.org/10.7554/eLife.47990.026

*Figure 2—figure supplement 1d*) had no effect on the rate of incorporation of exogenous palmitate into cellular PCs in BMDMs (*Figure 5c*).

It has been proposed that the Kennedy pathway is coupled to endogenous cellular fatty acid synthesis (*Ecker et al., 2010*; *Ridgway and Lagace, 2003*). In accordance, acetate incorporation into cellular PCs and total lipids showed a similar fold decrease (approximately 30%) as the reduction in de novo PC synthesis rate in CCTα-null macrophages (*Figure 5d–e*). Overall, we found that reducing the rate of de novo PC synthesis in macrophages proportionally decreased the rate of incorporation of lipids derived from de novo lipogenesis, which are known to be incorporated by the Kennedy pathway, but did not affect the rate of exogenous palmitate incorporation into membrane lipids.

## Loss of *Pcyt1a* in macrophages increases membrane PUFA abundance that protects against palmitate-induced ER stress

As our tracer experiments could not explain the diminished ER stress response observed in CCTα-null BMDMs in response to palmitate, we performed global lipidomic analysis of CCTα-null and control BMDMs. As described previously (*Tian et al., 2008*), total quantities of PC and PE in macrophages were unaffected by *Pcyt1a* deletion (*Figure 6—figure supplement 1a*). Unexpectedly, CCTα-null macrophages showed an enrichment in PUFA-containing PC levels compared to controls (*Figure 6a*, *Figure 6—source data 1*). Consistent with such observation, the expression of sterol regulatory element-binding protein 1 (SREBP1) target genes, which are known to be downregulated by high levels of PUFA-containing PLs in the ER (*Hagen et al., 2010*), was lower in CCTα-null macrophages than controls (*Figure 6—figure supplement 2*). Furthermore, changes in PE composition were similar to qualitative PC changes in CCTα-null and control cells (*Figure 6—figure supplement 1b*), indicating that reducing de novo PC biosynthesis rate promotes PUFA accumulation in membrane PLs.

We observed similar levels of saturated PC species between CCTα-null and control BMDMs under basal conditions (*Figure 6b*). Interestingly, the increased abundance of PUFA-containing PCs in CCTα-null BMDMs was at the expense of decreased mono- and diunsaturated PC species (*Figure 6a–b*). We confirmed these findings by analysing total BMDM fatty acid composition, which showed increased relative abundance of arachidonic (20:4n6), docosapentaenoic (22:5n6) and docosahexaenoic acids (22:6n3), while palmitoleic and oleic acid levels were reduced in CCTα-null cells compared to controls (*Figure 6c*).

As expected, palmitate treatment caused a large increase in the abundance of saturated PC species in BMDMs (*Figure 6—figure supplement 3b*). Interestingly, in palmitate-treated macrophages, PC saturation was increased mostly at the expense of reduced mono- and diunsaturated PCs, while the proportion of 3 or more double bond-containing PC species was largely unaffected by palmitate treatment (*Figure 6b* and *Figure 6—figure supplement 3b*). Consequently, CCTα-null BMDMs showed higher membrane PUFA levels than controls even after prolonged treatment with palmitate, leading to diminished palmitate-induced PC saturation (*Figure 6b* and *Figure 6—figure supplement 4*). Prolonged palmitate treatment elevated PC:PE ratio to a similar extent in both CCTα-null and control BMDMs, suggesting that changes in total PC levels or PC:PE ratio were unlikely to explain differences in ER stress response between genotypes (*Figure 6—figure supplement 3a*).

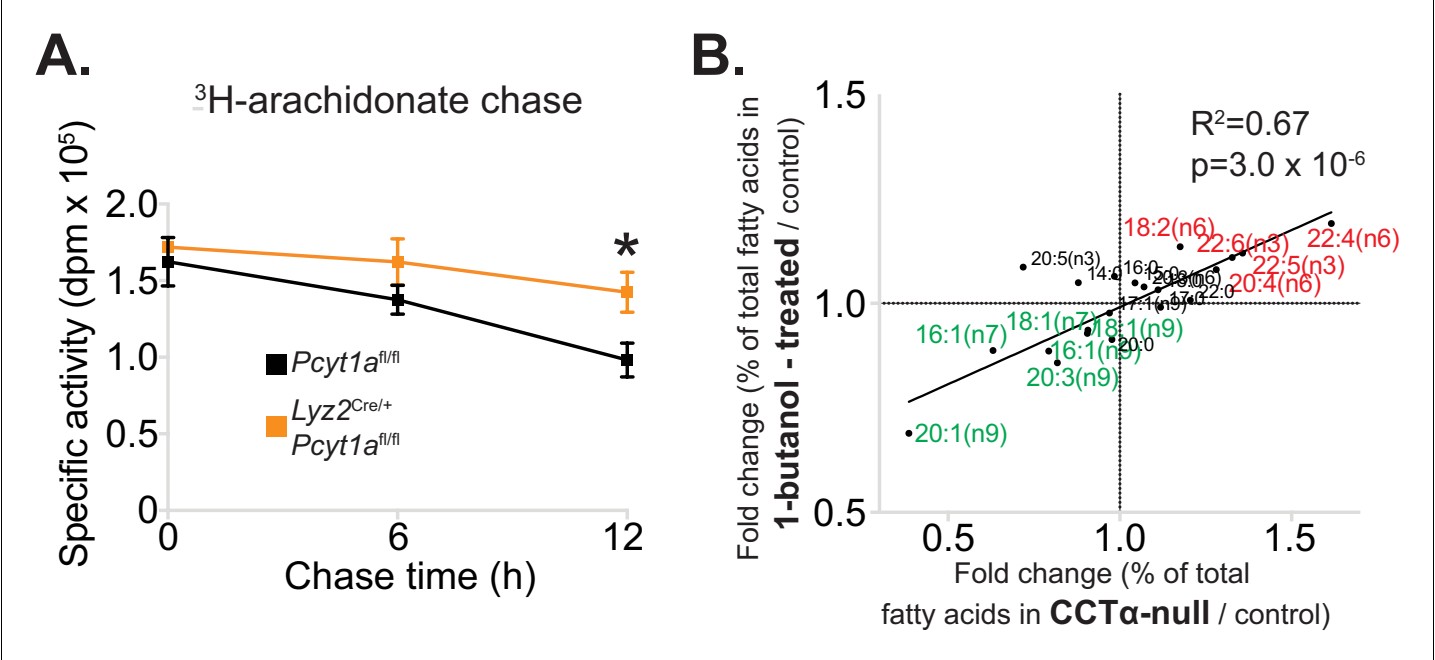

**Figure 7.** Reduced PC turnover increases membrane PUFA levels in macrophages. (**A**) [3]H-arachidonate levels in *Pcyt1a*[fl/fl] (n = 4) or *Pcyt1a*[fl/fl] *Lyz2*[Cre/+] (n = 4) BMDMs, pulsed with tracer amounts of [3]H-arachidonate for 16 hr and chased with medium for indicated periods of time. (**B**) Linear regression analysis of the correlation between fold changes in the molar fatty acid percentage induced by genetic *Pcyt1a* deletion (n = 4) and by PLD inhibition using 15 μM 1-butanol for 24 hr (n = 4).

DOI: https://doi.org/10.7554/eLife.47990.028

Our lipidomics analysis demonstrated that reduced CCTα activity could affect the levels of fatty acids other than palmitate and specifically resulted in the preferential accumulation of long chain PUFAs. These results were consistent with increased remodelling of PCs, most likely by LPCAT3. Furthermore, our lipid analysis also suggested an explanation for the lower ER stress in response to palmitate. Considerable literature has demonstrated that increased PUFA content in cellular membranes is protective against palmitate induced ER stress (*Rong et al., 2013*; *Yang et al., 2011*), suggesting a mechanistic explanation for the protective effects of *Pcyt1a* deletion against palmitate toxicity.

In order to experimentally demonstrate that CCTα-null macrophages exhibited lower ER stress in response to palmitate due to qualitative changes in membrane PL composition, we performed a rescue experiment of palmitate-treated BMDMs using exogenous arachidonic acid. Compared to palmitate treatment alone, 10:1 molar mixture of palmitate and arachidonate reduced *Tnf* and ER stress marker gene expression to the same basal level in both CCTα-null and control BMDMs (*Figure 6d*), suggesting that elevated PUFA levels negate the inflammatory effects of palmitate in macrophages lacking *Pcyt1a*. Overall, our results were consistent with a loss of CCTα resulting in a shift to a PUFA-rich membrane fatty acid composition that was protective against exogenous palmitate.

## Reduced PC turnover promotes membrane PUFA accumulation in *Pcyt1a*-deficient macrophages

Finally, we set out to explain how reduced CCTα activity could lead to an alteration in the fatty acid composition of PC. Our hypothesis was that the increased half-life of membrane PCs in CCTα-null macrophages might allow more time for PCs to be remodelled to contain PUFAs via the Lands cycle. CCTα-null macrophages have previously been shown to have reduced PC turnover rates without changes in total PC levels (*Tian et al., 2008*) and our experimental findings confirmed these results, as we have observed reduced de novo PC synthesis rate and unchanged PC abundance in CCTα-

null BMDMs compared to controls (*Figure 2—figure supplement 1d* and *Figure 6—figure supplement 1a*).

We next sought to confirm that the increased levels of PUFAs in CCTα-null macrophages were due to a lower turnover of PC. To do so, we performed a pulse-chase experiment using $^3$H-arachidonic acid. Indeed, CCTα-null BMDMs had increased retention of arachidonic acid in their membranes compared to control cells (*Figure 7a*). Overall, our results showed that the rate of PC turnover in macrophages is negatively associated with PUFA retention in PLs.

Importantly, while the data from *Pcyt1a*-deficient models demonstrated lower PC turnover, increased levels of long-chain PUFAs and lower PUFA turnover rates, this data came from congenic models lacking *Pcyt1a*. To exclude this being a phenomenon unique to genetically manipulating *Pcyt1a*, we sought to manipulate PC turnover rates via an alternative route. To do so, we pharmacologically blocked PC hydrolysis by inhibiting phospholipase D activity using 1-butanol. In accordance with our hypothesis, reducing PC turnover by inhibiting PC conversion to phosphatidic acid phenocopied the effects of reduced CCTα activity on cellular membrane fatty acid composition (*Figure 7b*).

## Discussion

Here, we demonstrate for the first time that obesity is characterised by an increase in de novo PC synthesis pathway in ATMs. We show that the increase in ATM de novo PC biosynthesis during obesity is pathophysiological using a macrophage-specific genetic model of reduced CCTα activity. Reducing de novo PC synthesis rate in ATMs alleviates obesity-induced WAT inflammation and improves systemic glucose metabolism. Mechanistically, we show that decreasing CCTα activity in macrophages does not reduce PC levels, but instead leads to a compensatory reduction in PC degradation and maintenance of normal PC levels. Because of this reduced PC turnover, more time is afforded for PC remodelling enzymes to act on PCs, leading to an increase in PUFA-containing PC species that are protective against ER stress in response to palmitate. Our results reveal a novel relationship between the regulation of de novo PC synthesis, PC turnover and membrane saturation.

Our study extends on existing findings regarding the importance of CCTα in mediating SFA-induced lipotoxicity in cultured macrophages. Two recent independent reports have suggested that de novo PC synthesis is responsible for exogenous palmitate incorporation into membrane PCs (*Gianfrancesco et al., 2019*; *Robblee et al., 2016*). *Robblee et al. (2016)* based their conclusions on data obtained from pharmacologically inhibiting CCTα using miltefosine at a dose of 100 µM, which inhibits de novo PC synthesis rate to a level that also leads to a reduction in exogenous palmitate incorporation, a result we reproduce here (*Figure 5C*). However, lower doses of miltefosine reduce de novo PC synthesis rate without affecting palmitate incorporation into cell membranes. At 25 µM miltefosine, we detect a 30% reduction in de novo PC synthesis rate with no reduction in palmitate incorporation, which phenocopies our genetic model of *Pcyt1a* deficiency in BMDMs. Importantly, our data show that a 30% reduction in de novo PC synthesis rate is sufficient to ameliorate ER stress, highlighting the capacity for changes in PC biosynthetic rate to regulate ER stress in a manner that does not require changes in palmitate incorporation. In this publication, we describe such a mechanism; that of reduced CCTα activity leading to increased PC half-life and thus permitting the establishment of a membrane composition that is protective against palmitate-induced ER stress. Similarly, *Gianfrancesco et al. (2019)* have utilised siRNA-mediated *PCYT1A* knockdown in cultured human macrophages to achieve approximately 50% reduction in de novo PC synthesis rate and demonstrated that it reduces SFA-induced inflammation. Due to a lack of tracer experiment in their study, we believe that reduced IL-1b secretion in their *PCYT1A* knockdown model is due to increased membrane PUFA content, as our experimental findings demonstrate that at 50 µM miltefosine, we detect a 50% reduction in de novo PC synthesis rate with no significant reduction in palmitate incorporation.

Further support for a role of PC turnover in regulating ER stress comes from recent work investigating TLR4 signalling. It has recently been shown that palmitate is not a direct TLR4 agonist, but instead requires TLR4 activation-induced changes in intracellular metabolism in order to promote ER stress and inflammation in macrophages (*Lancaster et al., 2018*). Importantly, it has previously been demonstrated that TLR4 activation increases the rate of de novo PC synthesis and PC turnover in macrophages in order to provide a supply of new membranes for secretory vesicle formation in

Golgi apparatus (*Sanchez-Lopez et al., 2019*; *Snider et al., 2018*; *Tian et al., 2008*). As such, our findings are in line with a mechanism in which basal TLR4 activation increases sensitivity of cells to palmitate by increasing PC turnover. In support of this concept, we demonstrate that decreasing de novo PC synthesis protects macrophages from palmitate-driven ER stress and inflammation. Furthermore, two recent reports have demonstrated that TLR4 activation in macrophages increases the transcription of *Slc44a1*, encoding choline transporter CTL1 (*Sanchez-Lopez et al., 2019*; *Snider et al., 2018*). As we observed increased *Slc44a1* transcript levels in $Lep^{ob/ob}$ ATMs compared to controls, in future it will be of interest to test whether the increase in *Pcyt1a* transcript in ATMs isolated from obese mouse and human WAT is dependent on TLR4 activation.

While ATMs undergo pro-inflammatory activation during obesity, liver macrophages do not (*Morgantini et al., 2019*). It is likely that for this reason we observed an increased de novo PC synthesis pathway activity in ATMs, but not in liver macrophages isolated from $Lep^{ob/ob}$ mice. While speculative, the absence of pro-inflammatory activation and normal de novo PC synthesis rate in liver macrophages could explain why hepatic genes related to metabolism and inflammation were comparable between CCTα-null $Lep^{ob/ob}$ BMT and control mice.

The observed effect size of macrophage-specific *Pcyt1a* deletion on systemic insulin sensitivity on the $Lep^{ob/ob}$ genetic background is relatively small. This could potentially be explained by two factors: 1) relatively small decrease (30%) in de novo PC synthesis rate in $Lyz2^{Cre/+} Pcyt1a^{fl/fl}$ macrophages, due to the poor penetrance of $Lyz2^{Cre/+}$ on *Pcyt1a* allele, as described previously (*Zhang et al., 2000*); 2) Macrophage-specific *Pcyt1a* deletion having a stronger impact on white adipose tissue insulin sensitivity, compared to muscle and liver. In either case, our data demonstrates a link between the increase in de novo PC biosynthesis rate in ATMs and the development of adipose tissue inflammation and insulin resistance.

Finally, inactivating mutations in *PCYT1A* gene have been linked to several human pathologies, including retinal dystrophy, spondylometaphyseal dysplasia and lipodystrophy (*Hoover-Fong et al., 2014*; *Payne et al., 2014*; *Yamamoto et al., 2014*). Our results suggest that besides controlling the production of bulk cellular PC mass, CCTα activity can affect the fatty acid composition of cell membranes by regulating their turnover, thus potentially explaining why homozygous *PCYT1A* mutations manifest in specific tissue disorders, and not in a systemic failure of proliferating cells. Such specificity is also reflected in our CCTα-null $Lep^{ob/ob}$ BMT model, where *Pcyt1a* deletion altered the inflammatory profile of ATMs, but not liver macrophages. In future, it would be interesting to study the relative impact of *Pcyt1a* deletion on different cell types, as well as same cell types present in different tissue microenvironments.

# Materials and methods

## Key resources table

| Reagent type (species) or resource | Designation | Source or reference | Identifiers | Additional information |
|---|---|---|---|---|
| Genetic reagent (*M. musculus*, male) | $Lep^{ob/ob}$ | The Jackson Laboratory | IMSR Cat# JAX:000632, RRID:IMSR_JAX:000632 | |
| Genetic reagent (*M. musculus*) | $Lyz2^{Cre/+}$ | *Clausen et al., 1999* | IMSR Cat# JAX:004781, RRID:IMSR_JAX:004781 | $Lyz2^{tm1(cre)Ifo}$; Donated by Dr. Susan Jackowski |
| Genetic reagent (*M. musculus*) | $Pcyt1a^{fl/fl}$ | *Zhang et al., 2000* | IMSR Cat# JAX:008397, RRID:IMSR_JAX:008397 | $Pcyt1a^{tm1Irt}$; Donated by Dr. Susan Jackowski |
| Antibody | anti-CD45 | BD Biosciences | 564279, RRID:AB_2651134 | Flow cytometry (1:200) |
| Antibody | anti-CD11b | BD Biosciences | 564443, RRID:AB_2722548 | Flow cytometry (1:200) |
| Antibody | anti-SiglecF | BD Biosciences | 562757, RRID:AB_2687994 | Flow cytometry (1:200) |
| Antibody | anti-F4/80 | BioLegend | 123116, RRID:AB_893481 | Flow cytometry (1:200) |

*Continued on next page*

*Continued*

| Reagent type (species) or resource | Designation | Source or reference | Identifiers | Additional information |
|---|---|---|---|---|
| Antibody | Anti-F4/80 (IHC) | Bio-Rad | Cl:A3-1, RRID:AB_1102558 | IHC(1:100) |
| Antibody | anti-CD206 | BioLegend | 141723, RRID:AB_2562445 | Flow cytometry (1:200) |
| Antibody | anti-CD11c | BioLegend | 117336, RRID:AB_2565268 | Flow cytometry (1:200) |
| Antibody | anti-Ly6G | BioLegend | 127608, RRID:AB_1186099 | Flow cytometry (1:200) |
| Antibody | anti-CD301 | BioLegend | 145704, RRID:AB_2561961 | Flow cytometry (1:200) |
| Antibody | anti-CCTa | Abcam | ab109263, RRID:AB_10859965 | WB(1:1000 dilution in 3% milk TBS-T), 4°Covernight |
| Antibody | anti-beta Actin | Abcam | ab8227, RRID:AB_2305186 | WB(1:1000 dilution in 3% milk TBS-T), 4°C overnight |
| Antibody | anti-P-Thr183/Tyr185 JNK | Cell signalling | 9251, RRID:AB_331659 | WB(1:1000 dilution in 3% BSA TBS-T), 4°C overnight |
| Antibody | anti-JNK | Cell signalling | 9252, RRID:AB_2250373 | WB(1:1000 dilution in 3% BSA TBS-T), 4°C overnight |
| Antibody | anti-P-Thr180/Tyr182 p38 MAPK | Cell signalling | 4511L, RRID:AB_2139679 | WB(1:1000 dilution in 3% BSA TBS-T), 4°C overnight |
| Antibody | anti-p38 MAPK | Cell signalling | 9212, RRID:AB_330713 | WB(1:1000 dilution in 3% BSA TBS-T), 4°C overnight |
| Antibody | Anti-P-Ser473 AKT | Cell signalling | 4060, RRID:AB_2315049 | WB(1:1000 dilution in 3% BSA TBS-T), 4°C overnight |
| Antibody | Anti-P-Thr308 AKT | Cell signalling | 9275, RRID:AB_329828 | WB(1:1000 dilution in 3% BSA TBS-T), 4°C overnight |
| Antibody | Anti-pan AKT | Cell signalling | 9272, RRID:AB_329827 | WB(1:1000 dilution in 3% BSA TBS-T), 4°C overnight |
| Antibody | Anti-GAPDH | Cell signalling | 97166, RRID:AB_2756824 | WB(1:1000 dilution in 3% BSA TBS-T), 4°C overnight |
| Sequence-based reagent | List of nucleotide sequences | This paper | PCR primers | See Table S3 for the full list of qPCR primer sequences |
| Commercial assay or kit | Alpha Trak two glucose meter | Zoetis | N/A | |
| Commercial assay or kit | TruSeq Stranded mRNA Library Prep (96 Samples) | Illumina | 20020595 | |
| Commercial assay or kit | Vybrant Phagocytosis Assay Kit | Thermofisher | V-6694 | |
| Commercial assay or kit | RNeasy Mini Kit (250) | Qiagen | 74106 | |
| Chemical compound, drug | Insulin | Novo Nordisk | EU/1/02/230/003 | |

*Continued on next page*

Continued

| Reagent type (species) or resource | Designation | Source or reference | Identifiers | Additional information |
|---|---|---|---|---|
| Chemical compound, drug | STAT 60 | AMS Biotech | CS-502 | |
| Chemical compound, drug | Reverse Transcriptase M-MLV | Promega | M170b | |
| Chemical compound, drug | M-MLV RT 5x buffer | Promega | M351A | |
| Chemical compound, drug | MgCl2 25 mM | Promega | A351B | |
| Chemical compound, drug | Random Primers | Promega | C118A | |
| Chemical compound, drug | dNTP Mix 10 mM | Promega | U151B | |
| Chemical compound, drug | Chlorofom | Sigma | 34854 | |
| Chemical compound, drug | Methanol | Sigma | 34860 | |
| Chemical compound, drug | BF3 Methanol | Sigma | B1127 | |
| Chemical compound, drug | Hexane | Sigma | 34859 | |
| Chemical compound, drug | Ammonium formate | Sigma | 516961 | |
| Chemical compound, drug | Acetonitrile | VWR | 83640.320 | |
| Chemical compound, drug | 1-butanol | Sigma | 281549 | |
| Chemical compound, drug | Miltefosine | Sigma | M5571 | |
| Chemical compound, drug | Arachidonic acid | Cayman | 90010 | |
| Chemical compound, drug | Palmitic acid | Cayman | 1000627 | |
| Chemical compound, drug | [1-$^{14}$C]- palmitic acid | Perkin Elmer | NEC075H050UC | |
| Chemical compound, drug | [1,2-$^{14}$C]- acetic acid, sodium salt | Perkin Elmer | NEC553250UC | |
| Chemical compound, drug | Methyl-[$^{3}$H]- choline chloride | Perkin Elmer | NET109250UC | |
| Chemical compound, drug | [5,6,8,9,11,12, 14,15-$^{3}$H(N)]- arachidonic acid | Perkin Elmer | NET298Z050UC | |
| Chemical compound, drug | Bovine Serum Albumin | Sigma | A8806 | BSA used for cell culture experiments |
| Chemical compound, drug | Hionic-Fluor scintillation liquid | Perkin Elmer | 6013319 | |
| Chemical compound, drug | Opti-Fluor scintillation liquid | Perkin Elmer | 6013199 | |
| Chemical compound, drug | Ethanol | Sigma | 459836 | |
| Chemical compound, drug | DAKO Real Peroxidase Blocking solution | Agilent | S2023 | |
| Chemical compound, drug | MOM ImmPress Polymer Reagent | Vector | MP-2400 | |

*Continued*

| Reagent type (species) or resource | Designation | Source or reference | Identifiers | Additional information |
|---|---|---|---|---|
| Chemical compound, drug | DAB Peroxidase (HRP) Substrate Kit | Vector | SK-4100 | |
| Chemical compound, drug | Dako REAL Haematoxylin | Agilent | S2020 | |
| Software, algorithm | Metabolizer algorithm | *Cubuk et al., 2018* | n/a | http://metabolizer.babelomics.org |
| Software, algorithm | MassHunter Workstation Software Quantitative Analysis (Version B.07.00) | Agilent Technologies Inc | n/a | |
| Software, algorithm | Thermo Xcalibur Quan browser integration software (Version 3.0) | Thermofisher | n/a | |
| Software, algorithm | HALO AI | Indica Labs | n/a | |
| Software, algorithm | TopHat (Version 2.0.11) | *Kim et al., 2013* | RRID:SCR_013035 | |
| Software, algorithm | EdgeR | *Robinson et al., 2010* | RRID:SCR_012802 | |
| Software, algorithm | HiPathia | *Hidalgo et al., 2017* | n/a | http://hipathia.babelomics.org |

## Mice

This research has been regulated under the Animals (Scientific Procedures) Act 1986 Amendment Regulations 2012 following ethical review by the University of Cambridge Animal Welfare and Ethical Review Body (AWERB). Mice were housed 3–4 per cage in a temperature-controlled room (21°C) with a 12 hr light/dark cycle, with 'lights on' corresponding to six am. Animals had *ad-libitum* access to food and water. A standard chow diet (DS-105, Safe Diets) was administered to all animals from weaning, consisting of 64.3% carbohydrate, 22.4% protein and 13.3% lipid of total calories. Only male mice were used for in vivo experiments. Male and female mice (8–20 weeks of age) were used for in vitro BMDM cultures.

## Generation of myeloid cell-specific cctα KO mouse

Macrophage-specific *Pcyt1a* knockout mouse (CCTα mKO) was generated by crossing a mouse model containing loxP sequences surrounding *Pcyt1a* alleles (*Pcyt1a*$^{fl/fl}$) to the *Lyz2*$^{Cre/+}$ (*Clausen et al., 1999*) mouse. *Pcyt1a*$^{fl/fl}$ mouse was generated by Prof. Ira Tabas and Dr. Susan Jackowski as described (*Zhang et al., 2000*), and was gifted to us on a mixed C57Bl/6J, 129/Sv genetic background by Dr. Suzanne Jackowski. *Pcyt1a*$^{fl/fl}$ and *Lyz2*$^{Cre/+}$ lines were backcrossed to a C57Bl/6J genetic background using Marker-Assisted Accelerated Backcrossing (MAX-BAX, Charles River, UK) technology until SNP genotyping confirmed >99% background purity.

All experimental macrophage-specific knockout mice were produced by crossing *Lyz2*$^{+/+}$ with *Lyz2*$^{Cre/+}$ animals on a floxed/floxed background, yielding a 1:1 Mendelian ratio of control (floxed/floxed *Lyz2*$^{+/+}$) to knockout (floxed/floxed *Lyz2*$^{Cre/+}$) offspring.

## Bone marrow transplant

4–6 week-old WT or *Lep*$^{ob/ob}$ host mice for bone marrow transplant were purchased from Jackson laboratories and were allowed to acclimatise for at least 2 weeks before the experiment. At 8 weeks of cage, mice were split into two groups of equal average body weight and equal average fed blood glucose concentration (Respective BW and glucose values ± SEM for control and CCTα mKO BMT: 47.1 ± 0.78 g and 46.8+1.68 g, p=0.87; 27.9 ± 3.05 mM and 24,4+3.46 mM, p=0.45). All mice were given 1% Baytril antibiotic in drinking water a day before irradiation. All mice received two doses of 5.5 Gy of radiation using Caesium 60 source. Two hours post irradiation, donor bone marrow cells (10 million/mouse) were injected into the tail veins of the irradiated mice. The cells from one donor

mouse were used for up to two host mice. Host mice were then housed at 3–4/cage, with 1–2 mice carrying *Pcyt1a*<sup>fl/fl</sup> bone marrow and 1–2 - *Pcyt1a*<sup>fl/fl</sup> *Lyz2*<sup>Cre/+</sup> bone marrow in each cage. Mice were kept on 1% Baytril for 1 month, monitored and weighed regularly until 12 weeks of age. A single *Lep*<sup>ob/ob</sup> mouse carrying *Pcyt1a*<sup>fl/fl</sup> bone marrow had to be culled due to health reasons. Mice were then housed under standard housing conditions throughout the duration of the study.

## Cells

### Culture and differentiation of bone-marrow derived macrophages

Femur and tibia bones from mice were isolated and cleaned, and 10 ml of Roswell Park Memorial Institute Medium (RPMI)−1640 (Sigma) was flushed through each bone using a syringe. Bone marrow cells were counted manually, pelleted by centrifugation, and re suspended in RPMI-1640 with 20–30% L929 conditioned medium, 10% heat-inactivated FBS (Gibco, Thermofisher Scientific) and 100 U/ml penicillin- streptomycin (Thermofisher Scientific) (macrophage differentiation medium). To differentiate into macrophages, cells were seeded in 10 cm non-culture treated plates (Falcon) at a density of $5 \times 106$ cells per plate per 10 ml of macrophage differentiation medium and cultured for 7 days at 37°C in 5% $CO_2$. On day 5 of differentiation, medium was removed, and 10 ml of fresh macrophage differentiation medium was added to each plate. On day 7 of differentiation, macrophages were detached using ice-cold PBS containing 1 mM EDTA, counted using Countess automated cell counter (Thermofisher), centrifuged at 500 g, room temperature for 5 min and re suspended in macrophage differentiation medium at the concentration of $5 \times 10^5$ cells/ml. Immediately after, cells were plated for experiments at the following densities: 500 µl/well of 24-well plate, 1 ml/well of 12-well plate, 2 ml/well of 6-well plate and 10 ml per 10 cm plate. Cells were incubated for at least 24 hr after plating before conducting experiments.

To make L929 conditioned medium, L929 cells (CCL-1, ATCC) were seeded in DMEM supplemented with 10% heat-inactivated FBS, 100 U/ml penicillin-streptomycin and 2 mM L-glutamine (Sigma) at a density of 500,000 cells per 50 ml of medium per T175 tissue culture flask. Medium was harvested after 1 week of culture, and then 50 mL of fresh DMEM supplemented with 10% heat-inactivated FBS, 100 U/ml penicillin-streptomycin and 2 mM L-glutamine was added onto cells and harvested 1 week later. Batches obtained after the first and second weeks of culture were mixed at a 1:1 ratio, aliquoted and stored at −20°C.

### Isolation and culture of peritoneal macrophages

Immediately after sacrifice, 5 ml of PBS containing 3% FBS was injected into palpitate peritoneal cavity. As much liquid as possible was recovered, and the procedure was repeated two more times with fresh PBS containing 3% FBS. Pooled lavages were centrifuged at 400 g, 4°C for 5 min. Cells were resuspended in 1 ml DMEM supplemented with 10% heat-inactivated FBS, 100 U/ml penicillin-streptomycin and 2 mM L-glutamine (Sigma) and counted manually. The concentration was adjusted to $5 \times 10^5$ cells/ml, and cells were plated in 24-well plates with 500 µl of cell suspension/well. Medium was changed after 6 hr, and cell stimulations were performed the following day.

### Isolation of liver macrophages

Liver macrophages were isolated according to a previously published detailed method (*Aparicio-Vergara et al., 2017*). Briefly, livers of anesthetised mice were first perfused with calcium-free Hanks' balanced salt solution (HBSS), followed by collagenase digestion. After digestion, the hepatocytes were released by mechanical dissociation of the lobes and underwent several steps of filtration with calcium-containing HBSS and centrifugation at 50 g for 3 min. The supernatant containing non-parenchymal cells was loaded on a Percoll gradient (25% and 50%) and centrifuged for 30 min, at 2300 rpm, at 4°C. The interphase ring with enriched LMs was collected. The cells were then plated for 30 min and washed twice before RNA was extracted for subsequent analyses.

### Human adipose tissue biopsies

Subcutaneous adipose tissue biopsies were collected from 19 individuals undergoing bariatric bypass surgery, and adipose tissue macrophages were isolated. The metabolic parameters of individuals have been published earlier (*de Weijer et al., 2013*). The isolation of macrophages from these adipose tissue biopsies has been described and presented earlier (*Virtue et al., 2015*).

## Methods

### Glucose tolerance tests

Mice were fasted for 16 hr from 4 pm to eight am. Mice were single-housed at least 1 hr prior to being injected intraperitoneally with 1 mg/kg glucose. Blood samples for glucose measurement were taken at indicated times after the injection.

### Insulin tolerance tests

Mice were fasted for 6 hr from 8 am to two pm. Mice we single-housed at least 1 hr prior to being injected intraperitoneally with 0.75 IU/kg of human insulin. Insulin dose of 2.5 IU/kg was used for $Lep^{ob/ob}$ mice. Blood samples for glucose measurement were taken at indicated times after the injection.

### Isolation of stromal-vascular fraction (SVF) from WAT

Adipose tissues were removed after sacrifice, chopped thoroughly and resuspended in 10 ml digestion solution containing 7 ml Hanks' Balanced Salt Solution (HBSS, H9269, Sigma), 0.23 g bovine serum albumin (BSA, A8806, Sigma), and 20 mg collagenase type II (C6885, Sigma), filtered through 0.22 µm membrane. The digestion was performed at 37°C for 20 min, with horizontal shaking at 100 rpm. The digestion mixture was then passed through a 100 µm cell strainer (352360, Falcon) into a fresh tube and incubated at room temperature for 10 min, allowing the adipocyte fraction to layer on the surface. Adipocyte fraction was removed by pipetting. The remaining solution was centrifuged at 400 g, 4° for 5 min and pellet was re suspended in 1 ml of pre-cooled (at 4°C) FACS buffer (PBS, 1 mM EDTA, 3% heat-inactivated FBS). Total SVF cell number was determined by Countess automated cell counter (Thermofisher).

### Flow cytometry

SVF or BMDMs were collected and kept in FACS buffer (PBS, 1 mM EDTA, 3% HI-FBS) on ice. Cell were stained with LIVE/DEAD (Invitrogen) and non-specific binding was blocked with 5 µg/ml anti-CD16/32. Cell surfaces were then stained with anti-CD45, anti-CD11b, anti-Siglec-F, anti-F4/80, anti-CD301, anti-CD206, anti-CD11c. Cells were gated within the live single cell population as CD45+/Ly6g-/SiglecF-/CD11b+/F4/80+ for ATMs. For BMDMs, median fluorescence for indicated markers was measured in live single cell population. Data were acquired on LSRFortessa (BD Biosciences) using FACS Diva software and analysed with TreeStar FlowJo (Version vX0.7).

### Phagocytosis

Bacterial phagocytosis in BMDMs was measured by incubating cells for 2 hr with *E. coli* (K-12 strain) bacteria, labelled with the fluorescent dye fluorescein, according to manufacturer's protocol (Vybrant Phagocytosis Assay Kit, ThermoFisher).

### Radioisotope labelling

For radiolabelling experiments, radioisotope tracers were dissolved in macrophage differentiation medium at the following concentrations: 0.074 MBq/ml for methyl-[$^3$H] choline chloride (for PC synthesis assays), 0.06 MBq/ml for [1-$^{14}$C]-palmitic and [$^3$H]-arachidonic acids, and 0.148 MBq/ml for [1-$^{14}$C] acetic acid. BMDMs were plated at the standard densities in 24-well plates for PC synthesis and fatty acid incorporation/chase assays and in 6-well plates for lipid class analysis by thin-layer chromatography.

Radioactivity in total cells was determined by lysing cells in 100 µl of PBS containing 1% Triton X-100, adding the lysate to scintillation vial containing 5 ml of Hionic-Fluor scintillation liquid and subjecting it to liquid scintillation counting (LSC).

### Fatty acid treatments

All fatty acid treatments were done using FFAs conjugated to BSA (fatty acid and endotoxin free, A8806, Sigma). The conjugation was performed by preparing a sterile-filtered 5% BSA solution in macrophage differentiation medium. Both 5% BSA medium and concentrated fatty acid solution (100 mM of fatty acid in ethanol) were heated at 60°C before adding fatty acid solution dropwise

into 5% BSA medium in order to make a medium containing 2.5 mM fatty acid and 5% BSA (approximately 10:3 fatty acid to BSA molar ratio). This medium was then sonicated until it became completely clear and used as a stock solution for stimulations on the same day without sterile filtering. FFA-free BSA solution with equivalent amount of ethanol was used as a control. In dose-response experiments, the amount of BSA and ethanol in each condition was adjusted to the highest dose of palmitate. Unless otherwise indicated, fatty acid treatments were performed overnight for 16 hr.

## Extraction and quantification of RNA

RNA from cells was extracted using RNeasy Plus Mini kit (74106, Qiagen) following manufacturers' instructions. 30 µl of RNAse-free water was used for elution.

RNA from tissues was harvested by adding 1 ml of RNA Stat-60 reagent (Tel Test) to approximately 100 mg of frozen tissue placed in a Lysing Matrix D tube (MP Biomedicals). Samples were homogenised using a FastPrep homogeniser (MP Biomedicals) for $2 \times 45$ s at 5.5 m/s and centrifuged at 14,000 g for 5 min to pellet debris. The aqueous phase was transferred to a fresh tube containing 200 µl chloroform. Samples were mixed and centrifuged at 14,000 g, 4°C for 15 min. The clear upper phase containing RNA was removed and precipitated by mixing it with 500 µl isopropanol and incubating at room temperature for 10 min. Samples were centrifuged at 14,000 g, 4°C for 10 min and supernatants were discarded. RNA pellets were then washed with 70% ethanol, air-dried and re-suspended in 100 µl of RNAse-free water.

RNA concentration and purity were determined using Nanodrop ND-1000 spectrophotometer (Thermofisher Scientific). The absorbance was measured at 260 nm against RNAse-free water. A single A260 unit was assumed to be equal to 40 µg/mL of RNA. All RNA samples were stored at −80°C for subsequent processing.

## Quantitative real-time polymerase chain reaction (qRT-PCR)

Complementary DNA (cDNA) was generated using Promega reagents in a 20 µl reaction as follows: 500 ng RNA was added to 1 x M-MLV reverse transcriptase master mix (M351A) with 2.5 mM MgCl2 (A351B), 1.25 mM dNTPs (U151B), and 5 µg/mL random hexamers (C118A), and denatured at 65°C for 5 min before being transferred directly to ice in order to prevent the reassembly of the secondary structures of RNA. After the addition of 1 µL of M-MLV reverse transcriptase (M170b), the reaction was incubated at 37°C for 1 hr for cDNA synthesis and 95°C for 5 min for enzyme denaturation. cDNA was diluted 75-fold in RNAse-free water and stored at −20°C.

qRT-PCR was performed in a 13 µL reaction with 5 µl of diluted cDNA, 6.5 µl of 2x TaqMan or SYBR Green reagent (Applied Biosystems), 1.3 µl of 3 mM forward and reverse primer mix (including 1.5 mM of probe for TaqMan reactions) and 0.2 µl of RNAse-free water according to the default manufacturer's protocol (Applied Biosystems). Primer sequences are listed in *Supplementary file 3*. Reactions were run in duplicate for each sample and quantified using the ABI Prism 7900 sequence detection system (Applied Biosystems). Duplicates were checked for reproducibility, and then averaged; 'no reverse transcriptase' controls were included to check for genomic DNA contamination, and 'no template' controls were included to check for the formation of primer dimers. Product specificity was determined using a dissociation curve for SYBR green reactions. A standard curve generated from a pool of all cDNA samples was used for quantification. The expression of genes of interest was normalised using BestKeeper method to the geometric average of 3–4 housekeeping genes (for mouse: *18 s*, *36b4* and *Tbp*; for human: *Actb*, *B2m*, *Gapdh* and *Polr2a*), and data was expressed as arbitrary units or normalised to the average of control group.

## Whole adipose tissue RNA sequencing and analysis

2 µg of eWAT RNA was used to generate barcoded sequencing libraries using TruSeq Stranded mRNA Library Preparation Kit (Illumina) following manufacturer's instructions. After adjusting for concentration, the sequencing libraries were combined into 96-plex pools. The pooled libraries were sequenced on 3 lanes of an Illumina HiSeq 4000 instrument at single-end 50 bp (SE50), yielding an average of 15.7 million reads per sample. Library preparation was performed by the Genomics and Transcriptomic Core at the Institute of Metabolic Science. The sequencing was performed at the Genomics Core, Cancer Research UK Cambridge Institute.

RNA sequencing data was aligned using TopHat (V2.0.11) to the mouse GRCm38 genome and genes were counted using HTseq-count (V0.8.0) by the Genomics and Transcriptomic Core at the Institute of Metabolic Science. Data normalisation and differential gene expression analysis was performed with edgeR using TMM and generalised linear model methods, respectively. Pathway activity analysis was performed using HiPathia algorithm.

## Extraction of RNA, RNA sequencing and analysis of liver macrophages

RNA extraction was performed using the TRIzol Reagent according to the manufacturer's instructions (15596018, ThermoFisher).

RNA libraries were prepared using TruSeq Stranded mRNA kit (Illumina). The concentration of indexed libraries was quantified by RT–qPCR using the Universal Kapa Library Quantification Kit (KAPA Biosystems). Final libraries were normalised and sequenced on an Illumina HiSeq 2000 sequencer.

Raw fastq-files were aligned against the murine genome version mm10 using TopHat (v2.0.13) with all default options. BAM files containing the alignment results were sorted according to the mapping position. mRNA quantification was performed using FeatureCounts from the Subread package against the GRCm38-gencode transcripts database version seven (gencode.vM7.annotation.gtf) and the GRCh38-genocode transcripts database version 24 (gencode.v24.annotation.gtf) to obtain read counts for each individual Ensembl gene. The read count table of the dataset was normalised separately using DESeq2.

## Protein extraction and quantification

BMDMs were treated as described in legend, washed once with ice-cold PBS and lysed in ice-cold RIPA buffer (50 mM Tris-HCl, 150 mM NaCl, 1 mM EDTA, 0.1% SDS, 0.5% sodium deoxycholate, 1% NP-40, pH 7.4) containing Pierce protease and phosphatase inhibitors (88668, Thermofisher Scientific). 150 µl of RIPA buffer was used to lyse $10^6$ cells. Frozen white adipose tissue, liver and gastrocnemius muscle samples (approximately 50 mg) were ground in liquid nitrogen using mortar and pestle, and powdered tissue was then dissolved in 200 µl ice-cold RIPA buffer. Lysates were collected and centrifuged at 14,000 g, 4°C for 10 min to remove cell debris.

Protein concentration was determined by DC Protein assay (5000111, Biorad) according to manufacturer's instructions.

## Western blotting

Protein lysates were diluted in NuPAGE LDS sample buffer (NP0007, Thermofisher Scientific) containing 2.5% 2-mercaptoethanol and boiled at 95°C for 5 min. 10 µg of protein was then separated by electrophoresis using NuPAGE SDS-polyacrylamide gels (Thermofisher Scientific) and transferred to nitrocellulose membranes using the iBlot Dry Blotting System (Thermofisher Scientific). Membranes were blocked for 1 hr in 5% fat-free milk (Marvel) or 5% BSA in Tris-buffered saline containing 0.05% Tween (TBST) at room temperature and incubated overnight at 4°C with the appropriate primary antibody. Bound primary antibodies were detected using peroxidase-coupled secondary anti-rabbit antibody (7074, Cell signalling) and enhanced chemiluminescence (WBLUF0500, Millipore). Blots were exposed digitally using the ChemiDoc MP System (Bio-Rad), and bands were quantified using Image Lab software (Bio-Rad). The expression of proteins was normalised to a housekeeping protein (β-actin), and the phosphorylation status was determined by normalising to a respective total protein. All protein quantification data is expressed as arbitrary units.

## Adipose tissue histology and imaging

Adipose tissue samples for histology were placed in 10% formalin overnight, then transferred to 70% ethanol before embedding in paraffin. Different 4 µm sections were obtained from FFPE blocks and extra-coated with paraffin to preserve tissue integrity.

After incubating overnight at 37°C, sections were dewaxed using xylene and 100% industrial methylated spirits, then washed under running water for 5 min and kept in TBST. The sections were stained as follows: 1) blocking endogenous peroxidases for 5 min (DAKO Real Peroxidase Blocking solution, S2023); 2) wash in TBST; 3) blocking using serum for 20 min; 4) primary anti-F4/80 antibody incubation for 60 min; 5) wash in TBST for 5 min; 6) 30 min incubation with MOM ImmPress Polymer

Reagent (MP-2400); 7) wash in TBST; 8) DAB solution (5–10 min) prepared according to the manufacturer's instruction (DAB Peroxidase substrate kit, SK-4100); 9) wash in TBST; 10) 1 min incubation with Dako REAL Haematoxylin (S2020). The sections where then washed in tap water, dehydrated in graded alcohols, cleared in xylene and mounted.

All eWAT slides were scanned using a Zeiss Axio Scan Z1 and analysed using HALO software (Indica Labs, Corrales, NM). The 'tissue classifier module', utilising a state-of-the-art machine learning algorithm to identify tissue types based on colour, texture, and contextual features, was used to distinguish areas containing F4/80-positive cells (marked in dark green). 'Vacuole Quantification module' was then applied to analyse the adipocytes in dark green areas and in a whole section. Intact vacuoles completely surrounded by F4/80-positive cells were considered as CLS, while whole section vacuole analysis was used to determine average adipocyte area. The analyses were performed on the whole section to avoid selection bias; tissue edges were excluded using manual annotation. Halo was 'trained-by-example' on randomly selected images, and then the analysis was extended on the whole batch of sections with HALO automated pipeline.

## Lipid extraction

Total lipids from cells were extracted using a modified Folch extraction method. Glass pipettes were used throughout the procedure in order to avoid plastic-bound lipid contamination. 1 ml of HPCL-grade chloroform: methanol 2:1 v/v mixture was added to cell samples in a glass vial. Where applicable, appropriate amounts (calculated by approximating the average abundance of every fatty acid within the sample and adding matching amounts of standard) of 1,2-diundecanoyl-sn-glycero-3-phosphocholine (phospholipid standard, 850330C, Sigma) were included in extraction mixture as internal standard. Samples were homogenised by vortexing for 15 s. 200 µl of HPLC-grade water was added to each sample before vortexing for 2 min and centrifuging at 4000 g for 10 min. 700 µl of the lower lipid fraction was transferred to a 7 ml glass tube. A second extraction was performed by adding 700 µl of fresh HPLC-grade chloroform followed by vortexing and centrifugation as above. 900 µl of lower lipid fraction was collected and pooled with the first 700 µl fraction (total 1600 µl). Collected lipid fractions were dried under nitrogen stream. Dried lipids were stored at −20°C for subsequent processing, or resuspended in 100 µl chloroform, transferred to scintillation vials containing 5 ml of Opti-Fluor scintillation liquid (6013199, Perkin Elmer) and subjected to LSC.

## Thin-layer chromatography

BMDMs were treated in 6-well plates as indicated in legend and lipids were labelled as described. Lipids from cells were extracted and solubilised in 50 µl of HPLC-grade chloroform. 20 µl of lipids were then spotted at the bottom of 20 cm x 20 cm thin layer chromatography (TLC) silica plates (Z292974, Sigma). TLC plates were placed into hermetic glass chambers containing 250 ml of 65:25:4 chloroform: methanol: ammonium hydroxide v/v solution for phospholipid separation. Plates were allowed to develop until the solvent front was approximately 2 cm below the top of the plate. Plates were dried under laminar flow and incubated with radiographic films (47410, Fujifilm) in the dark for 1–3 days at room temperature. Radiographic films were developed using automated film developer and scanned. ImageJ software (NIH) was used to calculate the density of the bands on scanned radiograms.

## LC-MS lipid analysis

To the previously dried lipid samples, 60 µL of the lipid internal standard was added (methanol containing $CE(18:0)_{d6}$, $Ceramide(16:0)_{d31}$, $FA(15:0)_{d29}$, $LPC(14:0)_{d42}$, $PA(34:1)_{d31}$, $PC(34:1)_{d31}$, $PE(34:1)_{d31}$, $PG(34:1)_{d31}$, $PI(34:1)_{d31}$, $PS(16:0)_{d62}$, $SM(16:0)_{d31}$, $TG(45:0)_{d29}$, and $TG(54:0)_{d35}$, all at 10 µg/mL. The samples were then thoroughly vortexed, then dried under a gently stream of nitrogen. The samples were then reconstituted by adding 740 µL of 4:1 mix of isopropanol and acetonitrile, respectively, and vortexed ensuring there was no undissolved material. The samples were then analysed by LC-MS analysis.

Chromatographic separation was achieved using Acquity UPLC CSH C18 (50 mm x 2.1 mm, 1.7 µm) LC column with a Shimadzu UPLC system (Shimadzu UK Limited, Wolverton, Milton Keynes). The column was maintained at 55°C with a flow rate of 0.5 mL/min. A binary mobile phase system was used with mobile phase A; 60:40 acetonitrile to water, respectively, with 10 mM ammonium

formate, and mobile phase B; 90:10 isopropanol to acetonitrile, respectively, with 10 mM ammonium formate. The gradient profile was as follows; at 0 min_40% mobile phase B, at 0.4 min_43% mobile phase B, at 0.45 min_50% mobile phase B, at 2.4 min_54% mobile phase B, at 2.45 min_70% mobile phase B, at 7 min_99% mobile phase B, at 8 min_99% mobile phase B, at 8.3 min_40% mobile phase B, at 10 min_40% mobile phase B.

Mass spectrometry detection was performed on an Exactive Orbitrap mass spectrometer (Thermo Scientific, Hemel Hempstead, UK) operating in positive/negative ion switching mode. Heated electrospray source was used, the sheath gas was set to 40 (arbitrary units), the aux gas set to 15 (arbitrary units) and the capillary temperature set to 300°C. The instrument was operated in full scan mode from m/z 150–1200 Da.

Data processing was completed using Thermo Xcalibur Quan browser integration software (Thermo Scientific, Hemel Hempstead, UK). The identification of the lipid species was determined by an MS-signal for the corresponding theoretically calculated m/z accurate mass found at the expected retention time. The semi-quantitation of the lipids was calculated by the integration of the analyte MS-signal relative to the lipid class internal standard concentration.

## Quantitative analysis of fatty acid methyl esters (FAMEs)

In order to derive FFAs and esterified fatty acids from complex lipids into FAMEs, 750 µl of HPLC-grade chloroform: methanol 1:1 v/v solution was added to previously dried lipids in 7 ml glass vials. 125 µl of 10% boron trifluoride in methanol (134821, Sigma) was then added into each vial. Vials were sealed and incubated in an oven at 80°C for 90 min in order to hydrolyse fatty acid-glycerol and fatty acid-cholesterol ester bonds and form FAMEs. Samples were allowed to cool, and 1 ml of HPLC-grade n-Hexane and 500 µl of HPLC-grade water were added. Samples were briefly vortexed and centrifuged at 2000 g using benchtop centrifuge. The upper organic layer was transferred into 2 ml gas chromatography glass vials and dried under nitrogen stream.

Gas chromatography-mass spectrometry was performed with Agilent 7890B gas chromatography system linked to Agilent 5977A mass spectrometer, using AS3000 auto sampler. A TR-FAME column (length: 30 m, inter diameter: 0.25 mm, film size: 0.25 µm, 260M142P, Thermofisher Scientific) was used with helium as carrier gas. Inlet temperature was set at 230°C. Dried FAME samples were resuspended in 100 µl HPLC-grade n-Hexane. 1 µl of this solution was injected for analysis. The oven programme used for separation was as follows: 100°C hold for 2 min, ramp at 25 °C/min to 150°C, ramp at 2.5 °C/min to 162°C and hold for 3.8 min, ramp at 4.5 °C/min to 173°C and hold for 5 min, ramp at 5 °C/min to 210°C, ramp at 40 °C/min to 230°C and hold for 0.5 min. Carrier gas flow was set to constant 1.5 ml/min. If the height of any FAME peaks exceeded $10^8$ units, sample was re-injected with 10:1 – 100:1 split ratio. Identification of FAME peaks was based on retention time and made by comparison with those in external standards (Food industry FAME mix, 35077, Restek).

Peak integration and quantification was performed using MassHunter Workstation Quantitative Analysis software (version B.07.00, Agilent). Specific high-abundance ions from total ion chromatogram were chosen to calculate each fatty acid peak. The values for each fatty acid were expressed in molar percentages by dividing the area of each peak by the sum of all peak areas for a given sample. This analysis accounted for differences in total lipid content between samples.

## Statistical analysis and graphical representation of data

All data from experiments is represented as a mean, with error bars showing standard error of the mean and the number of replicates stated in legend. Some data is represented as a fold-change, and it is stated in legend to what value the data represented was normalised to generate the fold-change. Statistical tests used are also stated in legend. A student's t-test was used to compare two groups; one-way analysis of variance (ANOVA) was used to compare more than two groups, followed by Bonferonni's post-hoc test. Where more than one factor influenced the variable being measured, 2-way ANOVA was used to test for a significant effect of each factor as well as an interaction between factors.

All statistical tests were performed and graphs were generated using GraphPad Prism six software. Graphs and figures were edited for presentation using Adobe Illustrator CC 2015 software.

Metabolizer algorithm used to analyse microarray data can be accessed at http://metabolizer.babelomics.org and its methodology is presented in recent publications (*Çubuk et al., 2019*; *Cubuk et al., 2018*).

## Acknowledgements

*Pcyt1a*^fl/fl and *Lyz2*^Cre/+ mice were a kind gift from Dr. Susan Jackowski. We thank Daniel Hart, Sarah Grocott, Charley Beresford, Jade Bacon, Laura McKinven, Eerika Rasijeff and Agnes Lukasik for their excellent technical assistance in the animal work. All animal work was carried out in the Disease Model Core (MRC Metabolic Diseases Unit [MRC_MC_UU_12012/5]; Wellcome Trust Strategic Award [100574/Z/12/Z]). We also thank Brian Lam and Marcella Ma from the Genomics and Transcriptomics Core, James Warner from the Histology core and Gregory Strachan from the Imaging core for their technical assistance. All serum biochemistry was conducted by the Biochemistry Assay Lab (MRC Metabolic Diseases Unit [MRC_MC_UU_12012/5]). We thank the Wellcome Trust [102354/Z/13/Z], BHF [RG/18/7/33636], MRC [MC_UU_12012/2], Spanish Ministry of Economy and Competitiveness [SAF2017-88908-R], AstraZeneca through the ICMC (MA), the Swedish Research council (MA: 2015–03582), the Strategic Research Program in Diabetes at Karolinska Institutet (MA) and Mediq Tefa for funding this work. The research leading to these results has also received support from the Innovative Medicines Initiative Joint Undertaking under EMIF grant agreement n°115372, resources of which are composed of financial contribution from the European Union's Seventh Framework Programme (FP7/2007-2013) and EFPIA companies' in kind contribution.

## Additional information

### Funding

| Funder | Grant reference number | Author |
|---|---|---|
| Wellcome Trust | 4-year PhD programme in Metabolic and Cardiovascular Disease (102354/Z/13/Z) | Kasparas Petkevicius |
| British Heart Foundation | Programme Grant RG/18/7/33636 | Kasparas Petkevicius<br>Sam Virtue<br>Guillaume Bidault<br>Antonio Vidal-Puig |
| Wellcome Trust | Strategic Award | Kasparas Petkevicius<br>Sam Virtue<br>Guillaume Bidault<br>Antonio Vidal-Puig |
| Medical Research Council | MRC_MC_UU_12012/5 | Kasparas Petkevicius<br>Sam Virtue<br>Guillaume Bidault<br>Antonio Vidal-Puig |
| Medical Research Council | MRC_MC_UU_12012/2 | Kasparas Petkevicius<br>Sam Virtue<br>Guillaume Bidault<br>Antonio Vidal-Puig |
| Ministry of Economy and Competitiveness | SAF2017-88908-R | Cankut Çubuk<br>Joaquin Dopazo |
| Vetenskapsrådet | 2015-03582 | Cecilia Morgantini<br>Myriam Aouadi |
| Karolinska Institutet | MA Strategic research program in Diabetes | Cecilia Morgantini<br>Myriam Aouadi |
| BBSRC | BB/P028195/1 | Benjamin Jenkins<br>Albert Koulman |
| BBSRC | BB/M027252/2 | Benjamin Jenkins<br>Albert Koulman |
| Mediq Tefa | Unrestricted grant | Mireille J Serlie |

The funders had no role in study design, data collection and interpretation, or the decision to submit the work for publication.

**Author contributions**
Kasparas Petkevicius, Conceptualization, Data curation, Formal analysis, Funding acquisition, Investigation, Methodology, Writing—original draft, Writing—review and editing; Sam Virtue, Conceptualization, Data curation, Formal analysis, Supervision, Funding acquisition, Investigation, Methodology, Project administration, Writing—review and editing; Guillaume Bidault, Formal analysis, Investigation, Writing—review and editing; Benjamin Jenkins, Albert Koulman, Data curation, Formal analysis, Methodology, Conducted LC-MS lipid analysis; Cankut Çubuk, Joaquin Dopazo, Data curation, Software, Formal analysis, Methodology, Performed bioinformatics analyses of transcriptomic data; Cecilia Morgantini, Myriam Aouadi, Resources, Data curation, Formal analysis, Provided transcriptomic data and analysis of liver macrophages; Mireille J Serlie, Resources, Data curation, Formal analysis, Provided human adipose tissue samples and isolated human ATMs; Antonio Vidal-Puig, Conceptualization, Supervision, Funding acquisition, Investigation, Project administration, Writing—review and editing

**Author ORCIDs**
Kasparas Petkevicius (iD) https://orcid.org/0000-0003-2295-6065
Sam Virtue (iD) https://orcid.org/0000-0002-9545-5432
Cankut Çubuk (iD) https://orcid.org/0000-0003-4646-0849
Cecilia Morgantini (iD) https://orcid.org/0000-0003-3142-2508
Myriam Aouadi (iD) http://orcid.org/0000-0001-6256-7107
Antonio Vidal-Puig (iD) https://orcid.org/0000-0003-4220-9577

**Ethics**
Human subjects: Human samples used for this work had been generated as part of another study, that has already been published and is referenced in our manuscript (de Weijer et al, 2013). This study had been conducted to the highest ethical standards, and the ethics statement is available in the published paper.
Animal experimentation: All animal protocols were conducted in accordance with the UK Home Office and Cambridge University ethical guidelines.

**Decision letter and Author response**
Decision letter https://doi.org/10.7554/eLife.47990.038
Author response https://doi.org/10.7554/eLife.47990.039

## Additional files

**Supplementary files**
• Supplementary file 1. The list of biological processes increased in $Lep^{ob/ob}$ compared to WT ATMs at week 16 and no change at week 5, ranked in ascending order of adjusted p value.
DOI: https://doi.org/10.7554/eLife.47990.029

• Supplementary file 2. The list of differentially regulated GO biological processes in eWAT isolated from $Lep^{ob/ob}$ BMT $Pcyt1a^{fl/fl}$ and $Pcyt1a^{fl/fl}Lyz2^{Cre/+}$ mice, ranked in ascending order of p value. Supplementary file 2b. The list of differentially expressed genes in eWAT isolated from $Lep^{ob/ob}$ BMT $Pcyt1a^{fl/fl}$ and $Pcyt1a^{fl/fl}Lyz2^{Cre/+}$ mice, ranked in ascending order of p value.
DOI: https://doi.org/10.7554/eLife.47990.030

• Supplementary file 3. The list of qPCR primer sequences used in this publication. FAM/TAMRA reporter and quencher detection system was used for genes with indicated probe sequences, and SYBR was used for the remaining genes.
DOI: https://doi.org/10.7554/eLife.47990.031

• Transparent reporting form

DOI: https://doi.org/10.7554/eLife.47990.032

## Data availability

We are submitting raw source data Excel file for LC-MS lipidomics of Pcyt1a-deficient BMDMs, in both palmitate-treated and basal states (containing peak areas for each lipid species normalized to peak areas of respective internal standards) as Figure 6—source data 1. ATM microarray dataset (GSE36669) used in Figure 1 is already published and referenced in this manuscript. We are also submitting a list of differentially expressed genes detected by RNAseq in the eWAT of ob/ob bone marrow transplant mice, with a log (Fold change), log (CPM) and p-value indicated for each gene as Supplementary file 2b. We have uploaded raw RNA sequencing data of liver macrophages isolated from WT and ob/ob mice in the NCBI database under the following accession number: PRJNA541224.

The following dataset was generated:

| Author(s) | Year | Dataset title | Dataset URL | Database and Identifier |
|---|---|---|---|---|
| Morgantini C, Aouadi M | 2019 | Ob/ob liver macrophage RNA sequencing | https://www.ncbi.nlm.nih.gov/bioproject/PRJNA541224/ | NCBI BioProject, PRJNA541224 |

The following previously published dataset was used:

| Author(s) | Year | Dataset title | Dataset URL | Database and Identifier |
|---|---|---|---|---|
| Montaner D, Vidal-Puig A, Prieur X, Dopazo J | 2012 | Differential lipid partitioning between adipocytes and tissue macrophages modulates macrophage lipotoxicity and M2/M1 polarization in obese mice | https://www.ncbi.nlm.nih.gov/geo/query/acc.cgi?acc=GSE36669 | NCBI Gene Expression Omnibus, GSE36669 |

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
