## [Decision Letter]

Thank you for submitting your article "Accelerated phosphatidylcholine turnover in macrophages promotes adipose tissue inflammation in obesity" for consideration by *eLife*. Your article has been reviewed by two peer reviewers, one of whom is a member of our Board of Reviewing Editors, and the evaluation has been overseen by Satyajit Rath as the Senior Editor. The reviewers have opted to remain anonymous.

The reviewers have discussed the reviews with one another and the Reviewing Editor has drafted this decision to help you prepare a revised submission.

The reviewers agree that your study showing that increased turnover of phosphatidylcholine in macrophages has a role in adipose tissue inflammation is novel and important for the field. This opens a new path for exploration of an important topic. The data provided substantiate this conclusion.

However, there is one issue that was raised that deserves your attention and some revision in the manuscript prior to acceptance. The effects shown on systemic metabolism are very small (Figure 2) in this ob/ob mouse model, raising the question of whether there is a major impact on insulin resistance. It would be helpful if experiments were also conducted in the HFD mouse model which may show larger effects. Alternatively, if you have data on p-AKT blots to indicate the extent to which insulin signaling is effected in the ob model, including it would enhance the clarification of this issue, If such data are available, please include, *or* please add additional text in the Discussion directed to the importance of further investigation of this topic in the future.

---

## [Author Response]

There is one issue that was raised that deserves your attention and some revision in the manuscript prior to acceptance. The effects shown on systemic metabolism are very small (Figure 2) in this ob/ob mouse model, raising the question of whether there is a major impact on insulin resistance. It would be helpful if experiments were also conducted in the HFD mouse model which may show larger effects. Alternatively, if you have data on p-AKT blots to indicate the extent to which insulin signaling is effected in the ob model, including it would enhance the clarification of this issue, If such data are available, please include, or please add additional text in the Discussion directed to the importance of further investigation of this topic in the future.

We thank the reviewers for expressing their interest in our study and their positive feedback.

We agree that the effect sizes observed in ob/ob mice are small. In response to reviewers suggestion, we have produced p-AKT Western blots of gastrocnemius muscles and livers isolated from ob/ob mice carrying LysM-Cre Pcyt1a^fl/fl^ and control bone marrow and included them in the revised manuscript (Figure 3—figure supplement 3). These new data show that loss of *Pcyt1a* in macrophages does not affect liver insulin AKT phosphorylation and, while it appears to have some effect on muscle insulin sensitivity, AKT phosphorylation is more variable than in WAT and does not reach statistical significance. These results are in good accordance with the GTT and ITT data.

We have also included additional discussion regarding the effect sizes in the revised manuscript, noting the relatively low penetrance of the *Pcyt1a* knockout in macrophages. Finally, we have linked the discussion between the specificity of the effects observed in our mouse model, in terms of *Pcyt1a* deletion having different effects on specific macrophage populations, to the phenotypes of *PCYT1A* human mutations, where despite it being expressed in almost every cell, loss of function *PCYT1A* mutations only impact certain tissues and organs. We proposed to study this topic further in future.